# Robust analysis of allele-specific copy number alterations from scRNA-seq data with XClone

Rongting Huang [1,10], Xianjie Huang [1,2,10], Yin Tong[3,4], Helen Y. N. Yan [3,4], Suet Yi Leung [3,4,5,6], Oliver Stegle[7,8] & Yuanhua Huang [1,2,9] ✉

Somatic copy number alterations (CNAs) are major mutations that contribute to the development and progression of various cancers. Despite a few computational methods proposed to detect CNAs from single-cell transcriptomic data, the technical sparsity of such data makes it challenging to identify allele-specific CNAs, particularly in complex clonal structures. In this study, we present a statistical method, XClone, that strengthens the signals of read depth and allelic imbalance by effective smoothing on cell neighborhood and gene coordinate graphs to detect haplotype-aware CNAs from scRNA-seq data. By applying XClone to multiple datasets with challenging compositions, we demonstrated its ability to robustly detect different types of allele-specific CNAs and potentially indicate whole genome duplication, therefore enabling the discovery of corresponding subclones and the dissection of their phenotypic impacts.

The copy number alterations (CNAs), including copy loss, gain, and copy-neutral loss-of-heterozygosity (CNLOH or interchangeably LOH hereafter), have been widely shown associations in various cancers[1,2]. CNAs have also been found predictive to clinical outcomes in patients and can reoccur in certain tumor types[3]. Accurate detection of CNAs is therefore crucial for studying and understanding the genetic factors that contribute to the development and progression of cancers. High-throughput sequencing is one of the most widely used techniques for studying CNAs in cancers initially in a bulk manner, e.g., in the pan-cancer analysis[4]. Soon after, single-cell sequencing emerged, providing a better understanding of CNAs and their clonal evolution. Recent advances in single-cell DNA-seq at both technology and computational levels, e.g., ACT[5] and CHISEL[6], have further enhanced the analysis of CNA heterogeneity and its potential association with clonal fitness[7].

Single-cell RNA-seq (scRNA-seq) technologies have further fueled the CNA analysis thanks to its high popularity and dual readout of transcriptome and genetic makeups. However, its high technical sparsity makes it challenging to accurately call CNAs at a single-cell resolution. Several methods have recently been developed to tackle these challenges. InferCNV is one of the first methods proposed for this purpose[8] and remains widely used, achieving reasonable accuracy probably thanks to its effective smoothing by a moving-average method and multiple denoising treatments. CopyKAT is another recently proposed method that leverages a polynomial dynamic programming model to smooth the transformed data along genomic coordinates, with an emphasis on the detection of diploid cells[9]. However, both InferCNV and CopyKAT do not consider allelic information and hence cannot detect copy-neutral loss of heterozygosity

[1]School of Biomedical Sciences, The University of Hong Kong, Hong Kong SAR, China. [2]Center for Translational Stem Cell Biology, Hong Kong Science and Technology Park, Hong Kong SAR, China. [3]Department of Pathology, School of Clinical Medicine, LKS Faculty of Medicine, The University of Hong Kong, Queen Mary Hospital, Hong Kong SAR, China. [4]Centre for Oncology and Immunology, Hong Kong Science Park, Hong Kong SAR, China. [5]The Jockey Club Centre for Clinical Innovation and Discovery, LKS Faculty of Medicine, The University of Hong Kong, Hong Kong SAR, China. [6]Centre for PanorOmic Sciences, LKS Faculty of Medicine, The University of Hong Kong, Hong Kong SAR, China. [7]Division of Computational Genomics and Systems Genetics, German Cancer Research Center (DKFZ), Heidelberg, Germany. [8]Genome Biology Unit, European Molecular Biology Laboratory, Heidelberg, Germany. [9]Department of Statistics and Actuarial Science, The University of Hong Kong, Hong Kong SAR, China. [10]These authors contributed equally: Rongting Huang, Xianjie Huang. ✉e-mail: yuanhua@hku.hk

(LOH). On the other hand, HoneyBADGER[10] and CaSpER[11] were proposed to account for the allelic shift signal for detecting a full spectrum of CNAs. However, given the coverage of a single variant is extremely low, their independent use of individual single nucleotide polymorphisms (SNPs) often suffers from high noise. Instead, Numbat was recently introduced to substantially resolve such challenge by aggregating multiple variants via population haplotype-based phasing[12], a strategy that has been proven powerful in scDNA-seq[6]. However, this method may sacrifice the resolution of clonal CNAs from a single-cell level to a pseudo-bulk level and its performance and robustness remain to be further assessed, especially on samples with extreme compositions of clonal CNAs.

In this work, to overcome these challenges, we introduce *XClone*, a statistical model to detect allele-specific CNAs on individual cells by modeling the raw read/UMI counts from scRNA-seq data. Briefly, this method accounts for two modules: the B-allele frequency (BAF) of heterozygous variants and the sequencing read depth ratio (RDR) of individual genes, respectively detecting the variation states on allelic imbalance and absolute copy number, which are further combined to generate the final CNA states (Fig. 1a, Supplementary Figs. 1–2, Methods). Uniquely, our method introduced two major innovations to boost the signal-to-noise ratio in detecting CNA states. First, in the BAF module, we implemented a three-step haplotyping, from individual SNPs to a gene by population-based phasing, from consecutive genes (100 by default) to a gene bin by an Expectation-Maximization algorithm, and from gene bins to a chromosome arm by a dynamic programming method. Importantly, we re-designed our original

genotyping software cellsnp-lite[13] to a tailored package *xcltk* to address the issue of double-counting Unique Molecular Identifiers (UMIs) that cover multiple nearby SNPs (see Methods). Second, we employed two orthogonal strategies to smooth the CNA-state assignments on both BAF and RDR modules: horizontally with hidden Markov models (HMM) along the genome coordinates and vertically with a k-nearest-neighbors (KNN) cell-cell connectivity graph, which not only denoises the data but also preserves the single-cell resolution. With these tailored designs, XClone is a powerful and robust tool for detecting allele-specific CNAs. Its high accuracy will be demonstrated through comprehensive benchmarking with four cancer data sets and one simulated data set that are from different sequencing platforms and have unique complexity features including distinct allele loss and balanced whole genome duplication.

## Results

### High-level description of XClone model and workflow

The primary goal of XClone is to identify the CNA state for each region in each cell, therefore this statistical task can be formulated as a mixture model that infers the assignment of each (matrix) element to the discrete CNA states. Here, as we focus on scRNA-seq data, the region unit is a gene by default (or a gene bin with 100 consecutive genes for the BAF module below). Specifically, XClone includes two primary independent modules: the BAF and the RDR modules, as illustrated in Fig. 1a. The BAF module is designed to consider allele-specific read counts originating from heterozygous variants, whereas the RDR module accounts for read counts related to the absolute number of

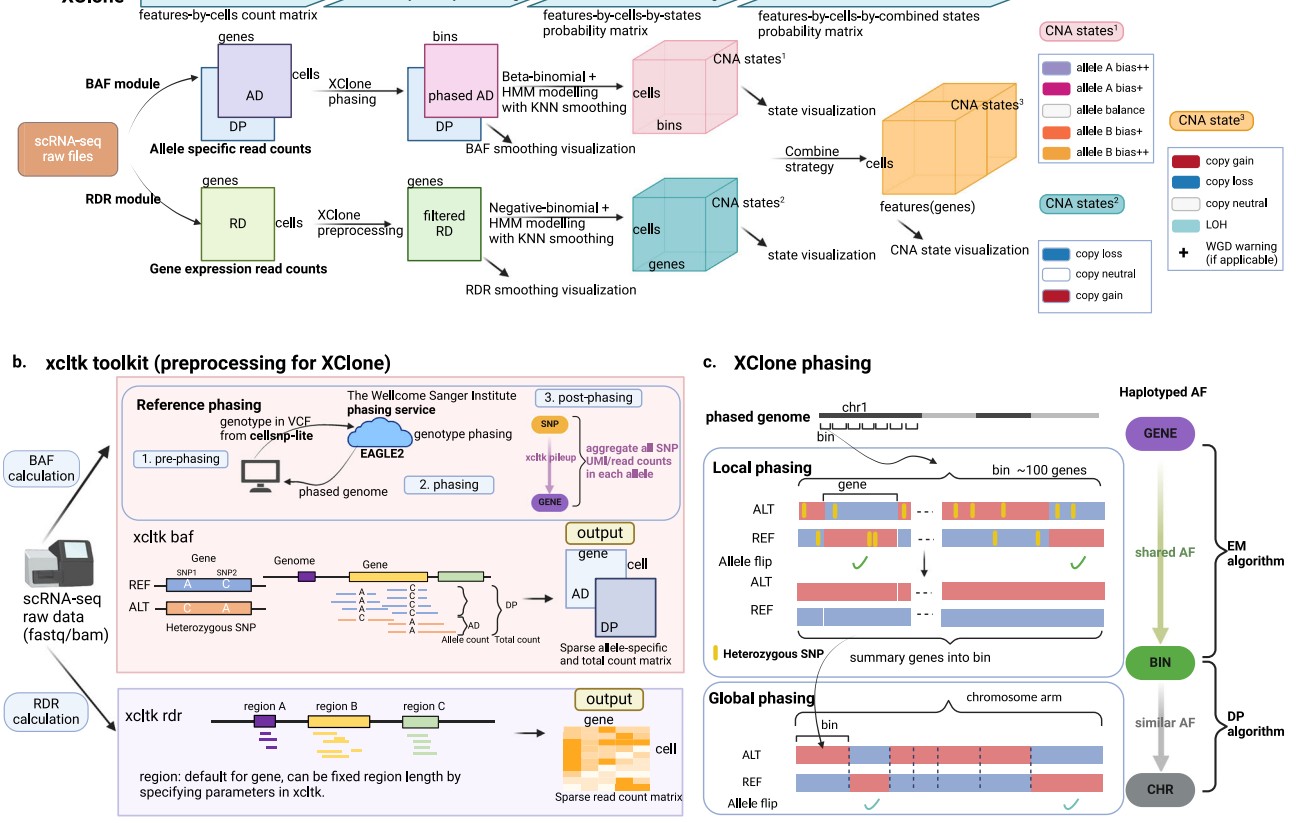

**Fig. 1 | Overview of XClone workflow. a** A step-by-step depiction of the XClone workflow. Starting with raw scRNA-seq files, XClone generates matrices of allele-specific (AD, DP) and gene expression read counts (RD), where features are organized by genomic positions and cells by assigned barcode order. The BAF and RDR modules are run independently in XClone, and their results are then combined to deduce the final CNA states. **b** A detailed representation of the quantification tool,

xcltk, which serves as a preprocessing step in XClone's CNA calling. **c** Phasing: Beyond the initial 'Reference phasing' step executed in xcltk preprocessing, XClone also conducts a 'Local phasing' and a 'Global phasing' step. Panel (**b**) created with BioRender.com released under a Creative Commons Attribution-NonCommercial-NoDerivs 4.0 International license.

copies reflected by gene expression level. With this setting, XClone has four major steps over two modules, for which sophisticated optimizations have been introduced to ensure high accuracy and robustness. Here, we describe the workflow over these two modules jointly for their shared preprocessing (Step 1) and CNA state combination (Step 4), while introducing the core parts of the BAF and RDR modules separately.

The XClone workflow starts with preprocessing both BAF and RDR modules, powered by our toolkit *xcltk* that is an upgradation from its base version cellsnp-lite[13] by tailored design (Fig. 1b, Section "Preprocessing: extract count matrices with xcltk"). Specifically, from the raw single-cell RNA sequencing (scRNA-seq) files, XClone constructs gene-by-cell matrices for allele-specific (AD, DP) counts and gene expression read counts (RD). These matrices are arranged such that the gene features are ordered by their genomic positions and cells by their assigned barcode order. Notably, our cellsnp-lite generally works well and has been used by a sister tool Numbat[12], while our upgraded xcltk resolves the fundamental issue of double counting UMIs when multiple SNPs are covered by the same UMI. Meanwhile, xcltk takes the genotypes phased from a population haplotype reference as input and automatically aggregates all heterozygous SNPs across a gene (see more details in Methods).

Within the BAF module, on top of reference based-phasing implemented in xcltk, XClone further employs two key phasing steps, namely Local and Global phasing respectively to aggregate allelic information from a gene to a gene bin (e.g., 100 genes by default) then to a chromosome arm, yielding the final phased allelic Depths (AD) and depth (DP) matrices in a shape of bins-by-cells (Fig. 1c). Subsequently, the AD and DP matrices undergo beta-binomial mixture modeling which is further smoothed by a Hiden Markov Model (HMM) along genome coordinates for the CNA state assignment. To enhance the quality of the mixture modeling, a KNN smoothing across cells in the transcriptome manifolds is also applied to the emission probability before inputting into the HMM framework. This process results in the generation of a bins-by-cells-by-states probability tensor, which is instrumental in identifying allele bias states in each bin and cell over five states (allele A bias ++/+, allele balance, and allele B bias +/++; Fig. 1a).

In parallel, the RDR module incorporates an initial preprocessing step in XClone to exclude less informative genes and cell-type markers (provided that cell-type information is available). This step is essential for refining the data and concentrating on the most critical components for the analysis. Following this, the expression matrix, structured as genes by cells, is subjected to negative-binomial statistical modeling paired with HMM iterations. This leads to the creation of a genes-by-cells-by-states probability tensor, which becomes the key signal in detecting gain or loss states in each gene and cell. Mirroring the BAF module, the RDR module also employs the KNN smoothing on the emission probability during the HMM process, returning three CNA states (copy loss, neutral, and gain).

In the final step, the bins-by-cells-by-states probability tensor, which represents allelic imbalance states, is mapped onto a gene scale. This tensor is then combined with the genes-by-cells-by-states probability tensor, which depicts absolute copy number variations. The result is a comprehensive genes-by-cells-by-states probability tensor that is used to detect states of copy gain, neutrality, loss, or loss of heterozygosity. This process is achieved through the combination strategy depicted in Supplementary Fig. 2. The probability tensor can be represented visually by selecting the state with the highest probability as the final result of CNA detection (over four states by default: copy loss, neutral, gain, and LOH). Of note, our combination also uniquely supports detecting balanced whole genome duplication via a warning function (see more in the astrocytoma data below).

In summary, XClone has advanced features through tailored preprocessing, sophisticated allele phasing, and effective two-view orthogonal smoothing, achieving reinforced signals on both BAF and RDR modules. Therefore, it enables robust and accurate detection of CNAs in scRNA-seq data, including those with complex structures. Moreover, besides the probabilistic assignment of CNA states, XClone also retains the concurrently calculated raw and smoothed BAF and RDR matrices for visualization purposes (Fig. 1a, Supplementary Figs. 3, 7, 9, 11). Notably, this allows us not only to double confirm whether XClone effectively enhances the signal-to-noise ratio for precise CNA detection but also to present the potential complex CNAs that are otherwise oversimplified by the four-state modeling. For more details of the XClone pipeline, we further provide a full illustration in Supplementary Figs. 1, 2 and detailed and modular methods in the Methods section.

## Assessment of CNA detection on a glioma with clone-specific allele loss

To evaluate the effectiveness of CNA detection, we utilized our XClone on a public tumor sample (glioma, BCH869) with a complex clonal structure that had been probed with high coverage using the Smart-seq2 platform. Notably, this sample has histone H3 lysine27-to-methionine mutations (H3K27M-glioma) and the dataset contains 489 malignant cells and 3 non-tumor cells[14]. As reported in the original study, a distinct subclonal structure has been observed in this population of cells by combining scRNA-seq (by using InferCNV) and whole-genome sequencing (WGS) data, where two major clones have different allele loss on chr14 (Fig. 2a). When applying XClone's RDR module, we indeed observed a copy loss on chr14 for almost all cells (Fig. 2b and Supplementary Fig. 3: smoothened log-transformed raw ratio visualization), consistent with that of InferCNV (Supplementary Fig. 4b). However, when leveraging the BAF module, XClone was able to identify the different allele loss on chromosome 14, matching the reported clones from WGS (Fig. 2c, Supplementary Fig. 3 BAF panel). Interestingly, XClone's BAF module detects medium but consistent allelic shifts on chr7 (clone 1), chr17q, and chr19p, endorsing the probable copy gain, instead of copy loss or LOH. In addition to the detection of these CNAs, XClone's combined mode provided further insight into the sample, identifying consensus CNAs such as copy gain on chr1q, chr2p, and chr17q. Clone-specific CNAs were also detected, including copy gain on chr7 and LOH on chr10 in clone 1, and copy gain on chr2q in both clones 1 and 3 (as shown in Fig. 2d). These results provide a more comprehensive understanding of the clonal structure of the sample and demonstrate the utility of XClone in detecting CNAs.

To compare the performance of XClone with other popular CNA detection methods, we applied InferCNV, CopyKAT, CaSpER, and Numbat to the same dataset (Fig. 2e and Supplementary Fig. 4). To evaluate the accuracy of these methods, we introduced a quantitative assessment metric using a receiver operating characteristic (ROC) curve at an entry level for each CNA state. This involved using the assignment probability (or transformed value) of each gene in each cell against the ground truth annotation to calculate the True Positive Rate and False Positive Rate when varying the threshold (Fig. 2f–h; Methods). The results showed that InferCNV and CopyKAT were effective at identifying absolute CN change (Supplementary Fig. 4) while Numbat showed potential in detecting allele-specific CNAs, including the major copy gains and LOH. However, Numbat had a notable flaw in copy loss detection, as it missed one of the two allele losses on chr14 (Fig. 2e). In contrast, XClone was able to identify nearly all CNAs across clones, achieving remarkable improvement on AUROC for all types of CNAs compared to any single counterpart method. Specifically, for copy gain, XClone achieved an AUROC of 0.925 compared to 0.853 by InferCNV and 0.845 by Numbat (as shown in Fig. 2f). For copy loss, XClone achieved an AUROC of 0.986 compared to 0.947 by InferCNV and 0.724 by Numbat (as shown in Fig. 2g). For LOH, XClone achieved an AUROC of 0.992 compared to 0.994 by Numbat, while InferCNV does not support this CNA type due to the lack of allelic information

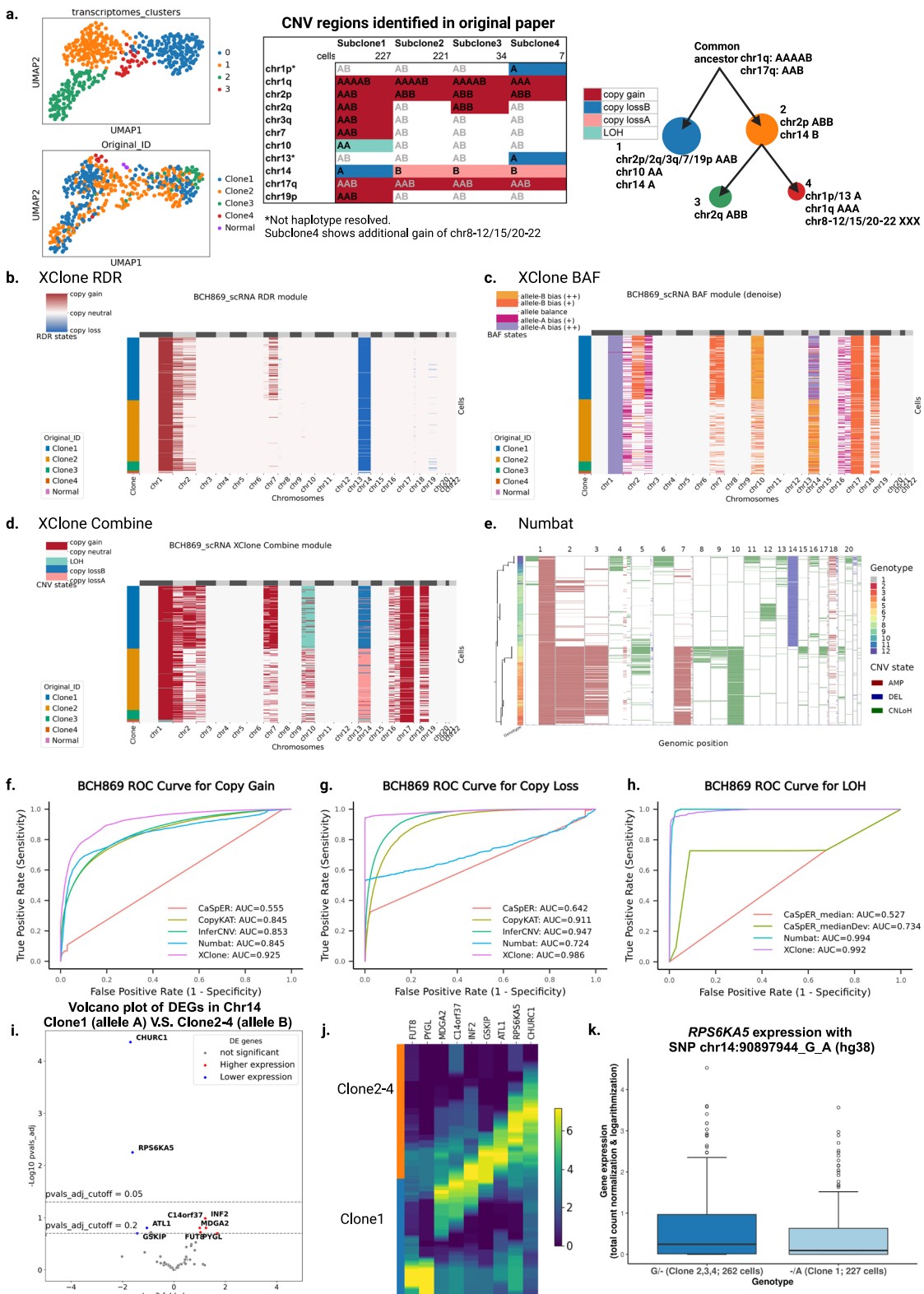

(Fig. 2h). Similar improvements were also observed when using the chromosome arm as a unit in the ROC curve (considering this resolution used by some methods; Supplementary Fig. 5) or performing a precision-recall curve (Supplementary Fig. 6). Notably, although XClone's plotting included cell cluster information, this information was not used in CNA estimation, similar to InferCNV. Thus, XClone preserved the CNA detection at the cell level rather than the cluster level.

Utilizing the information derived from allele-specific clonal analysis, we proceeded to perform a differential expression analysis between Clone 1 and Clones 2-4 of chromosome 14 within the BCH869 sample. This analysis revealed a set of differentially expressed genes (DEGs): 9 DEGs at an FDR < 0.2, and 2 DEGs at a more stringent FDR < 0.05, as shown in Fig. 2i, j. Interestingly, four out of the 9 DEGs are found to be cancer-related genes: *RPS6KA5*, *PYGL*, *FUT8*, and *INF2*. As an example, *RPS6KA5* exhibits

**Fig. 2 | CNA identification of a glioma sample BCH869 by XClone and performance comparison between methods. a** UMAP of BCH869 with different types of cell annotation: cell clustering by transcriptomes and CNA tumor clone ID identified by the original paper, CNA region identified for each clone and the inference clonal tree in the original paper. **b**–**d** The heatmaps show CNAs identified using XClone's RDR and BAF modules, and their combination, for BCH869. Each row in the heatmap represents a cell, and each column a genomic location: genes for RDR and combined heatmaps, and 100-gene bins for BAF heatmaps. Colors in the heatmap indicate the inferred copy number state at each location, reflecting genomic heterogeneity among cells. **e** presents a heatmap visualization of CNAs produced by Numbat. In this heatmap, each row typically represents a single cell, and each column represents a genomic location, nearly chromosome arm scale fragment. **f**–**h** Evaluation of the performance of benchmarked methods in identifying copy number gain, copy number loss, and loss of heterozygosity on the

BCH869 glioma sample (at the gene scale). **i** Volcano plot showing negative log10 *P*-values against average log2 fold changes (FC) for DEGs between Clone1 and Clone2-4 in chromosome 14. DEGs (Adjusted *p*-value < 0.05 or 0.2) are colored points above the horizontal dashed line. Significant DE genes expressed higher in Clone1 are highlighted in red, and lower in Clone1 in blue. **j** Obvious DEGs expression pattern in Clone1 and Clone 2-4. **k** *RPS6KA5* gene expression with SNP chr14:90897944_G_A. In the boxplot, the *x*-axis is the two groups with different genotypes of chr14:90897944 SNP: "G/-" in three clones 2, 3, 4 (262 cells); "-/A" in clone 1 (227 cells), where "-" stands for copy loss. The *y*-axis is the *RPS6KA5* gene expression that the normalized counts (by library size with scale to 10,000 followed by log1p transformation). The box center line, boundaries, and whiskers denote the medians, the first and third quartiles, and 1.5 times the interquartile range, respectively.

notable allele-specific expression on a SNP at chr14:9089 7944_G_A within its gene body, associating with the differences between Clone 1 and Clones 2-4 (Fig. 2k). The full allele-specific expressions on these 9 DEGs are shown in Supplementary Data 6. Therefore, XClone's accurate allelic CNA detection may offer potential insights into the dysregulation of cancer-related processes via a regulatory role.

## XClone identifies CNA states in an ATC sample with extremely low tumor purity

Next, to examine XClone's capability in analyzing droplet-based scRNA-seq (10× Genomics), we applied XClone to an anaplastic thyroid cancer sample (ATC2), covering 6224 cells. This sample was initially analyzed by CopyKAT[9], where the tumor purity was found extremely low in the original analysis and our replication (2%; Fig. 3a the second and fifth panels, respectively). Despite that the number of tumor cells is relatively small ($n = 123$), they are found to consist of two distinct sub-clones, which were initially observed by Numbat[12] and are readily captured by multiple other methods, including InferCNV, CopyKAT, and our XClone (Fig. 3a, bottom panels; Supplementary Fig. 8a). CaSpER failed to capture the same subclone signal in the ATC2 sample (Supplementary Fig. 8a, f).

However, the clonal CNA states detected by each tool varied dramatically, which may be attributed to the low tumor cell proportion. Notably, Numbat identified all CNAs as copy loss, which contrasts with the results of other tools. Although there was no ground truth from DNA assays, the BAF signal from scRNA-seq suggested that such widespread copy loss was a systematic error (Fig. 3e); for example, chr1p showed a strong signal of balanced B allele frequency, which indicated it should not be a copy loss. Additionally, lacking allelic information, InferCNV and CopyKAT cannot detect any CN-LoH, including chr1q, chr4q, and chr16. However, regarding absolute copy number, InferCNV and CopyKAT yielded patterns similar to those of XClone, including copy gains on chr5 (subclone 2 only), chr7, and 19, and copy loss on chr6p, concordant with commonly occurred CNAs in ATC patients[9,15]. These findings further validate the accuracy of our method in detecting CNAs with respect to both absolute copy numbers and allelic imbalance.

## XClone detects wide existence of LOH in a TNBC sample

Then, we applied XClone to a triple-negative breast cancer (TNBC) sample that was also assayed by droplet-based scRNA-seq (10× Genomics). As reported by the original study using CopyKAT, three clusters of cells were identified from the transcriptome, consisting of 300 normal cells and 797 tumor cells from two distinct CNA clones[9]. By re-analyzing this dataset, our RDR module confirms both consensus CNAs (chr1q, chr6p, chr8, chr12p, chr18) and subclonal CNAs including chr2q, chr4p, chr6q, chr7q, chr15p, chr16p in Clone 1 and chrX in Clone 2 (Fig. 4b, Supplementary Fig. 9). On the other hand, when applying XClone's BAF module, various regions were found with copy neutral in

absolute number but imbalanced allele frequency (Fig. 4c, d). This is further confirmed by XClone's combined mode, presenting a wide existence of copy-neutral LOH (Fig. 4e), which cannot be detected by methods omitting the allelic information, e.g., CopyKAT and InferCNV (Supplementary Fig. 10 b, c). One example is chr1p where strong allelic bias is observed by the smoothed BAF signal (Fig. 4c), further detected as LOH by XClone (with a minor probability of copy loss), while it is largely missed by Numbat (Fig. 4f). Specifically, consensus LOH includes chr1p and chr21 with high purity, and also chr11p, chr13, chr14, chr17, and chr19 with strong allele bias support, while clone 2 has specific LOH on chr3p and chr22.

Taking one step further, we leveraged the CNA probability detected by XClone to explore potential subclones within this sample, e.g., by K-means clustering on the flattened probability matrix here (XClone_ID in Fig. 4e; Methods). The results show that the clonal reconstruction by XClone is perfectly concordant with the original one (ARI metric: 0.965; Fig. 4a).

Based on the clone information, we conducted a differential expression analysis on TNBC1: Clone 1 versus Normal, and Clone 2 versus Normal; we found substantial numbers of DEGs from the CNA clones ($n = 4812$ and 4748 out of 18,892 genes for Clones 1 and 2 vs Normal, respectively; FDR < 0.05). Surprisingly, we found that the LOH regions return remarkably higher proportions of DEGs compared to non-LOH regions (Fig. 4g), both for Clone 1 (33.9% vs 21.4%; $p = 1.1e−21$; Fisher's exact test) and Clone 2 (27.8% vs 21.8%; $p = 1.7e-3$). Such preference implies a strong allele-specific expression, highlighting the capability of XClone to reveal unique regulatory-related insights, as these differences cannot be identified by methods that do not support allelic CNAs, like InferCNV and CopyKat. When examining the upregulated genes in Clone 1 (versus Normal), we found these genes are highly enriched cancer-related processes, particularly in the Wnt signaling pathway and cell-cell signaling by Wnt pathway (Fig. 4h), which are well-known oncogenic pathways, whose aberrant activation can lead to uncontrolled cell proliferation and inhibit cell death, promoting tumor growth including in breast cancer[16]. Similarly, upregulated genes in Clone 2 are also found cancer-related, for example via gland development (Fig. 4i), whose dysregulation links to breast cancer progression[17].

Furthermore, for the computational efficiency of the above two scRNA-seq datasets BCH869 and TNBC1, covering smart-seq with high coverage and 10x Genomics with high cell counts, CopyKAT is the most efficient method among all, closely followed XClone. While for methods using allelic information, XClone substantially reduced the running time (over 60%) compared to Numbat and CaSpER (Supplementary Data 4). Under default settings, XClone exhibits higher running efficiency in terms of user time and wall clock time, with acceptable memory usage. One additional advantage of utilizing XClone is its flexibility in selecting modules, namely the RDR module and BAF module, independently. This feature offers different options and choices to users based on their specific needs and preferences.

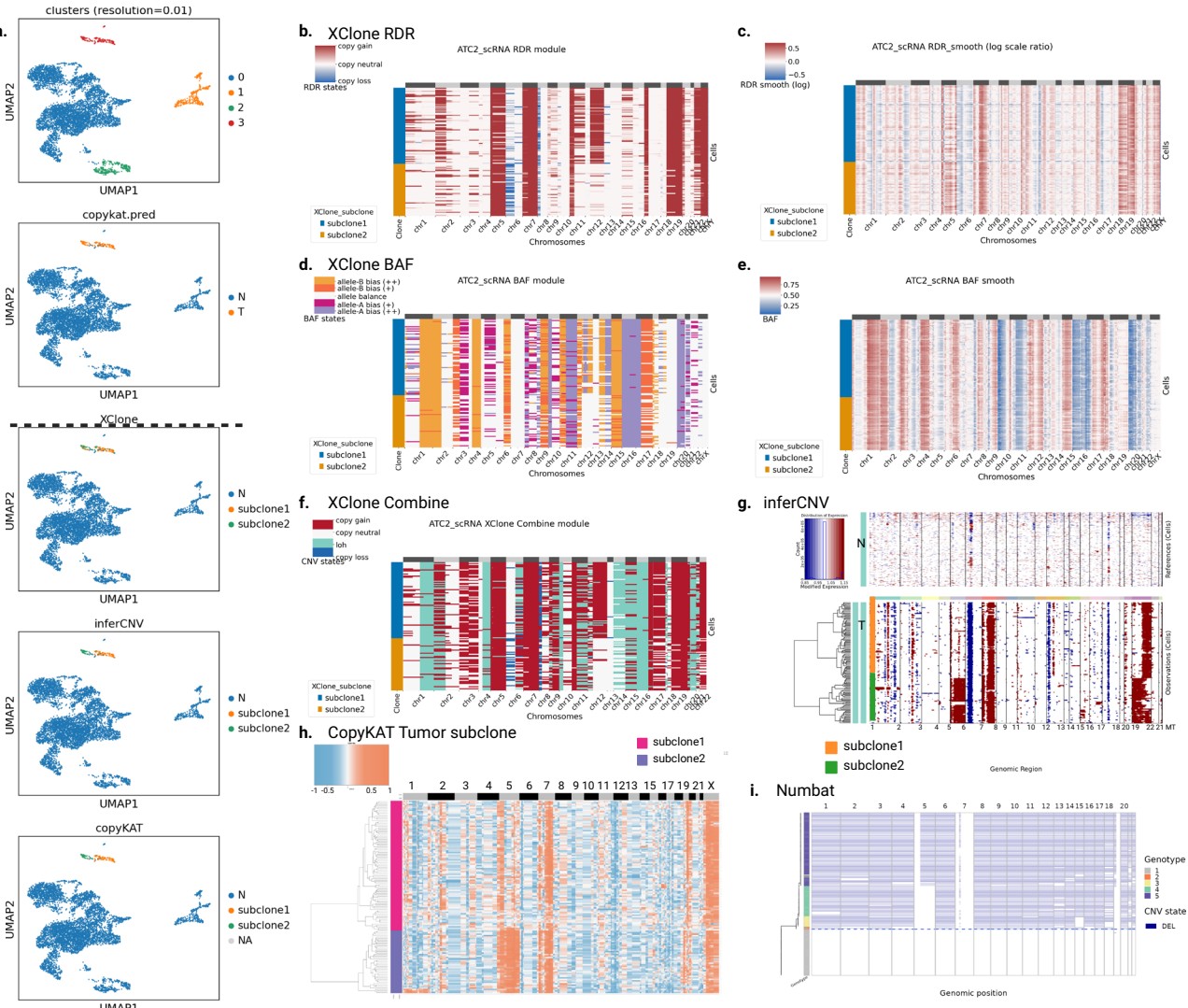

**Fig. 3 | CNAs identification of ATC2 sample by XClone and other methods (inferCNV, CopyKAT, Numbat). a** UMAP of ATC2 sample with different types of cell annotation: cell clustering by transcriptomes and Tumor/Normal annotation provided by CopyKAT at the top panel, tumor subclones identified by XClone, inferCNV and CopyKAT at the bottom panel, separated by a black dashed line. CNA heatmaps from the XClone RDR module demonstrate the detection of absolute copy number changes (copy gains or losses) in (**b**), alongside a smoothed visualization of the log-transformed raw ratios in (**c**). CNA heatmaps from the XClone BAF module are presented, showing allele imbalance detection in (**d**) and the smoothed B allele frequencies in (**e**). **f** Heatmap visualization of CNAs generated by XClone Combine module, indicating the final CNA detection results from the combination of both RDR and BAF modules. The CNA identification results from inferCNV (**g**), CopyKAT (**h**), and Numbat (**i**), respectively. Note, all heatmap figures (**b**–**i**; except InferCNV) only contain tumor cells. For the inclusion of all cells, refer to Supplementary Fig. 8.

A comprehensive analysis of the utilities associated with these methods is summarized in Supplementary Data 5.

## XClone infers complex CNA states with whole genome duplication in astrocytoma

Last, to validate that XClone effectively detects significant CNA clones accurately, we applied XClone to another droplet-based single-nuclei RNA-seq data (snRNA-seq) on an astrocytoma sample (a subtype of Glioblastoma Multiforme (GBM)), covering 4416 cells from 9 cell groups as annotated from the original study (see Fig. 5a). Uniquely, this is a second recurrent (2R) astrocytoma patient sample with *I*DH1(R132H) mutation, reported as a prominent case study in the newly developed scOne-seq technology (see the original study for more details about the sample)[18]. This sample therefore has been probed by multiple assays, including snRNA-seq (10× Genomics), bulk whole exome-seq (WES), and also the scOne-seq that profiles the DNA and RNA in the same cell in parallel with state-of-the-art efficiency,

making this dataset ideal for benchmarking. On the other hand, this sample has complex CNA profiles, as all tumor cells were reported with whole genome duplication on both alleles (4 copies in total) as the baseline. Therefore, it requires more care for interpreting the results, since all scRNA-seq-based methods cannot readily distinguish balanced duplication versus higher sequencing depth.

Here, our primary analysis focuses on the minor clone (2R clone 1, with 53 cells) that is newly emerged in the second recurrence (Fig. 5a), not only for its biological implications but also for its technical complexity. Biologically, all cells from this clone are classified alongside standard astrocytes from regular RNA data analysis, which implies a substantial degree of transcriptomic resemblance between 2R clone 1 and healthy astrocytes. The discovery of this minor clone with normal astrocyte phenotype is largely thanks to the joint profiling of transcriptome and genome by scOne-seq[18], otherwise it could potentially remain undiscovered. Technically, the CNA states of this clone are challenging to characterize, considering its whole genome duplication.

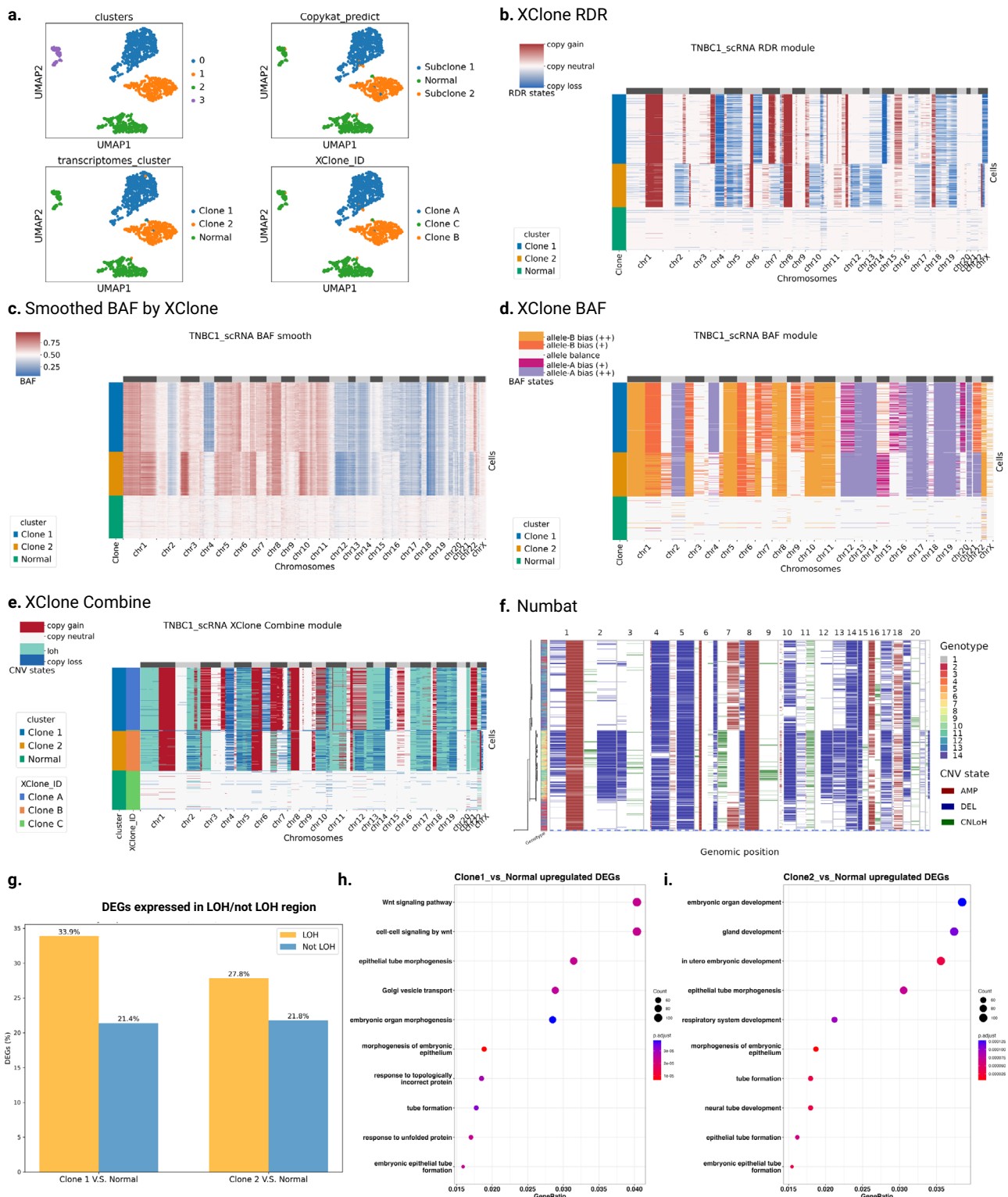

**Fig. 4 | TNBC1 CNAs identification by XClone and subclonal detection comparison with CopyKAT. a** UMAP of TNBC1 with different types of cell annotation. Transcriptomes cluster here is the annotation of cell clusters from expression and allele frequency combined analysis, which is used as annotation in analysis and visualization. **b–e** Heatmap visualization of CNAs generated by XClone RDR module, BAF smoothing (Same with Supplementary Fig. 9, Phased BAF after KNN smoothing and WMA smoothing), BAF module and combination of the 2 modules,

respectively for TNBC1, here the combined result shows that XClone detects similar subclones. **f** Heatmap visualization of CNAs generated by Numbat. **g** Comparison of the identified DEGs expressed in LOH and non-LOH regions. **h**, **i** Gene set enrichment analysis on the upregulated DEGs between Clone1 and Normal, Clone 2 and Normal, respectively, with default one-sided hypergeometric test from cluster-Profiler package (*p*-value cutoff of 0.05 and FDR cutoff of 0.2 (Benjamini–Hochberg method)).

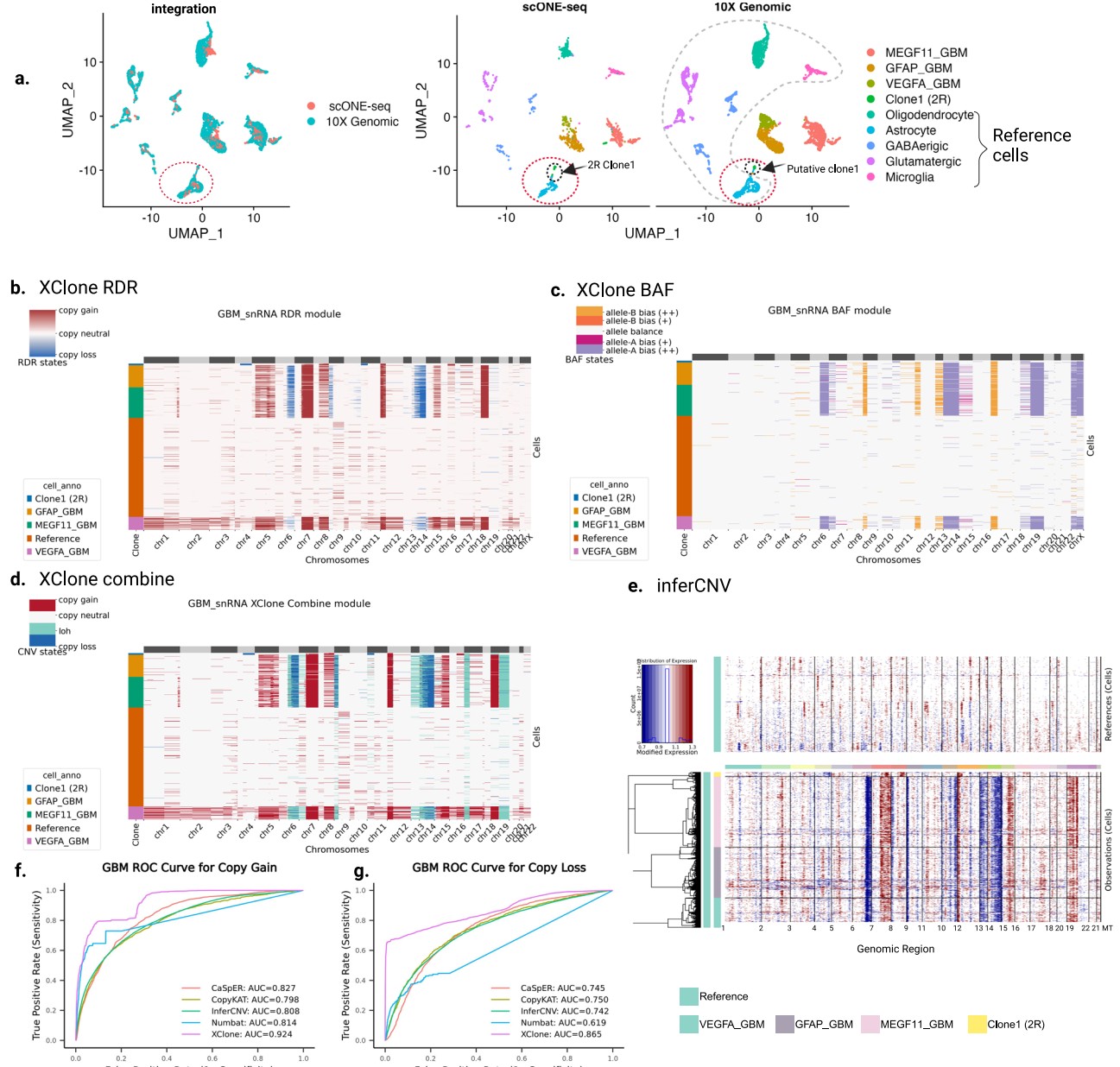

**Fig. 5 | CNA analysis of a second-recurrent (2R) astrocytoma sample snRNA-seq data, where XClone shows superior performance in detecting CNAs in the minor clone and indication of whole genome duplication. a** Transcriptome landscape of the IDH1-mutant astrocytoma. In the left panel, the UMAP shows the integration of scONE-seq and 10X Genomics snRNA-seq dataset. In the right panel, the split UMAP effectively illustrates the thorough intermingling of cell types identified independently by the two techniques. Notably, cells from the 2R clone1 (identified via scONE-seq) and the presumed clone1 cells (determined through 10X Genomics snRNA-seq) converge within the same integrated cluster (as demarcated by the black dashed circle). In the absence of genotype data provided by scONE-seq, the 2R clone1 is categorized as an astrocyte based solely on phenotype information (as demarcated by the red dashed circle). To ensure an equitable

comparison, we utilized the same set of cells as reference cells in all benchmarking methods (reference cells as demarcated by the grey dashed circle). **b–d** The heatmaps represent the visualization of CNAs generated by XClone's RDR module, BAF module, and the combination of these two modules, respectively for the 2R astrocytoma sample (10X Genomics snRNA-seq). The cell labels are matched with the annotations shown in (**a**) right panel. In this case, the integrated result demonstrates that XClone identifies evident subclone CNA profiles in 2R clone1. **e** The heatmap provides a visual representation of the CNAs generated by inferCNV. **f, g** Evaluation of the ability to detect copy number gains, copy number losses in the 2R Clone1 in astrocytoma sample. The ROCs provide a comparison of the performance of different methods, including the XClone, inferCNV, Numbat, CopyKAT, and CaSpER in identifying these key genomic alterations.

Therefore, when XClone's RDR module reports copy losses in chr4q and chr11p (Fig. 5b), the BAF module returns balanced allelic ratios (Fig. 5c) which should not happen with copy loss in a diploid-based genome. On the contrary, if seriously considered, these seemingly conflicting results imply the potential genome duplication, rather than erroneous results. Consequently, XClone introduces a warning system

for potential whole genome duplication, which is not supported by any other method, not to mention those without a BAF module.

To further highlight the challenge of detecting the CNAs on the minor clone (2R clone 1), we further computed the ROC curve at an entry level and compared the AUC between different methods. Here, by using the DNA profile from scOne-seq on this clone (the snRNA-seq

## Design of single-cell RNA-seq simulation for CNV detection benchmarking

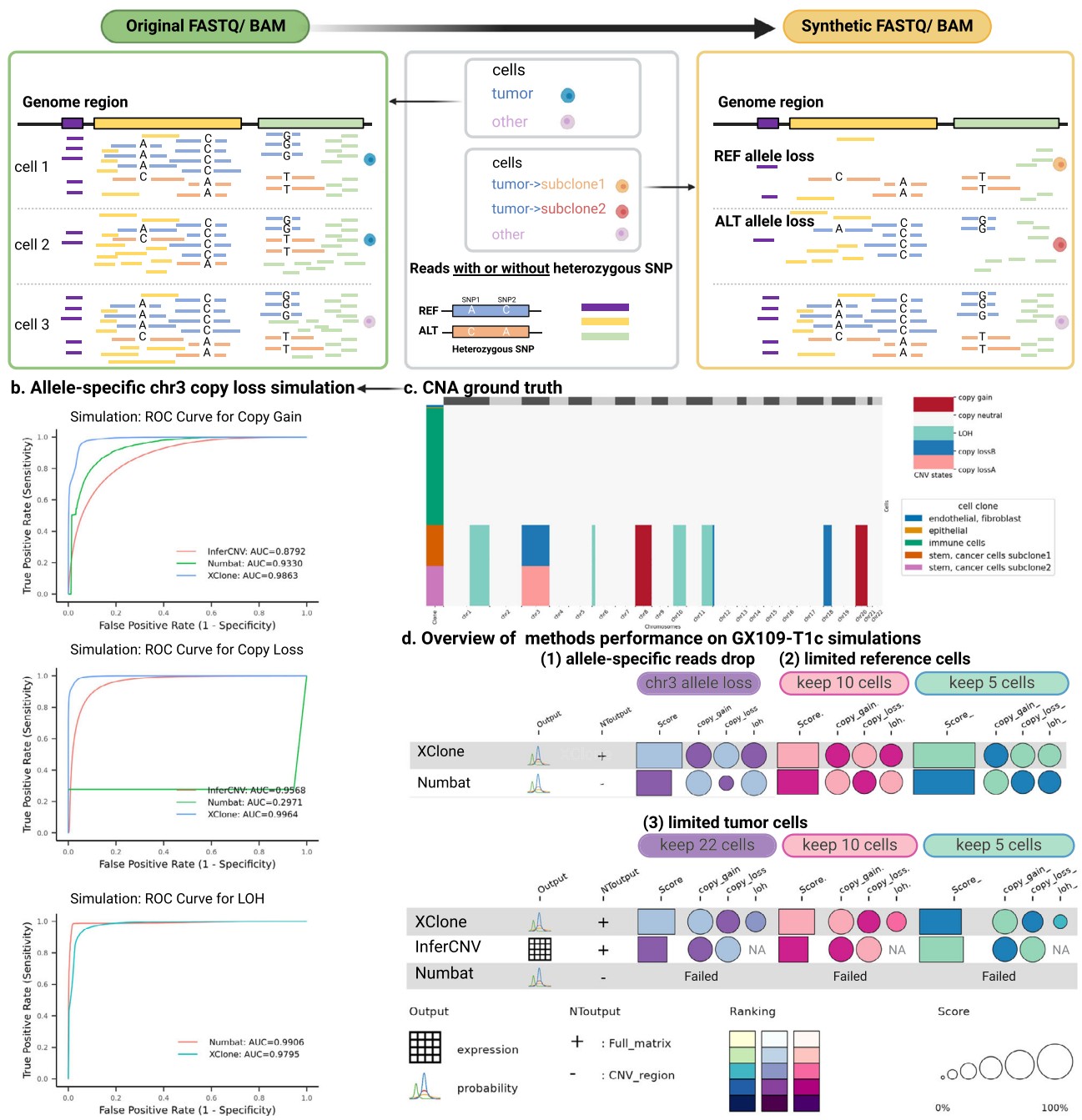

**Fig. 6 | Overview of the scRNA-seq simulation strategy for benchmarking CNA detection methods and the efficacy of these methods on the simulated datasets. a** A tailored simulator was developed to create scRNA-seq datasets that include allele-specific copy number loss within selected genomic regions. Take allele-specific copy loss on chromosome 3 in 2 subclones as an illustrative scenario: the left panel displays the original, unprocessed data from various cells spanning genomic regions. The right panel shows that tumor cell 1 is assigned to subclone 1, which exhibits a loss of the REF allele in certain regions of chromosome 3, while tumor cell 2 is part of subclone 2, characterized by a loss of the ALT allele. Cell 3 retains the initial, unaltered state. **b** Assessment of methods' ability to identify gene-scale copy number gains, copy number losses, and loss of heterozygosity (LOH) in the GX109-T1c simulation, specifically focusing on allele-specific copy loss on chromosome 3 in 2 subclones. **c** The corresponding CNA ground truth for allele-specific copy loss in different subclones. **d** A consolidated summary of the performance of various methods across distinct simulated scenarios. This includes a situation involving allele-specific copy loss in the two subclones and scenarios that only keep limited reference or tumor cells in datasets. The dot size denotes the score of AUROC in each CNA state and the bar size denotes the averaged score of loss, gain, and LOH. The color reflects the ranking in each item with a lighter color for better performance.

was aligned to the transcriptome of scOne-seq by integration), we obtained a single-cell level ground truth with copy gain on chr8, 12, 16, and 19, and copy loss on chr1, 4, 9, 11, 13, X. Of note, since all scRNA-seq-based methods cannot detect whole genome duplication, we kept

the copy loss/gain referring to the duplicated genome baseline. The results demonstrated that XClone reliably detected both copy gain and copy loss. XClone had the highest performance in detecting copy gain (AUC: 0.924), followed by CaSpER (AUC: 0.827), Numbat (AUC: 0.814),

InferCNV (AUC: 0.808) and CopyKAT (AUC: 0.798). In the detection of copy loss, XClone also demonstrated superior performance with an AUC of 0.865, followed by CopyKAT, CaSpER, and InferCNV (AUC ranging from 0.750 to 0.742). Numbat had an AUC as low as 0.619, partly due to the confusion of whole genome duplication (Fig. 5f, g). This result shows that XClone possesses the remarkable ability to detect small clones even in the early stages of the recurrence cancer sample.

Next, we also assessed the CNA profiles of the other three tumor cell groups: MEGF11_GBM, GFAP_GBM, and VEGFA_GBM, whose transcriptomes are distinct but CNA profiles are similar to the clone at the first recurrence, as reported by the original study[18], and also captured by XClone here (Fig. 5d). Interestingly, we observed complex patterns of CNAs in the "VEGFA_GBM" cells which are mesenchymal-like tumor cells located in an extreme hypoxia environment and hence may experience elevated chromosomal instability and dysregulated transcription[19].

### Allelic CNA simulations verify the superior robustness of XClone and provide a user guide

To further verify the superior performance of XClone, we aimed to conduct relevant simulations reflecting the complex scenarios we observed in the experimental datasets above. Considering that the phased allelic information is essential for our method (and also Numbat), we designed and implemented a reads-sampling-based simulator scCNASimulator (details of the simulator can be found in the Supplementary Technical Notes in Supplementary Information). Briefly, this tool is a Python package designed for CNA simulation in droplet-based scRNA-seq data. It mainly takes an indexed Binary Alignment Map (BAM) file and a clone-specific CNA profile as input and outputs a new indexed BAM file containing re-sampled reads with new tags (in UMIs) for the desired CNA alignments. Critically, when performing allele drop, it will use the genotyping phasing information to drop a specific allele (cartoon shown in Fig. 6a).

To simulate high-fidelity reads and CNAs, our scCNASimulator generally requires a reliable seed dataset (in BAM format). To achieve this, a fresh gastric cancer tissue was sequenced via 10x Genomics 5' scRNA-seq kit (GX109-T1c). This sample is an ideal seed dataset for the simulations: (1) the CNA clonal structure is simple with all micro-environment cells as CNA neutral and tumor cells as one CNA clone, (2) the CNA profile of the tumor cells is readily detectable and covers gain, loss and LOH, both of which are consistently supported by bulk WES, scDNA-seq and scRNA-seq data, all on this gastric cancer tissue sample (see details about this data in Methods section "Seed dataset GX109-T1c and the generation of BAM files for different simulations of challenging scenarios" and Supplementary Figs. 15–17). The CNA profile of the input seed dataset is also shown in Supplementary Fig. 18a, including copy gain at chr8, chr20, copy loss at chr12p (chr12:7288360-16757762) and chr18, and CN-neutral LOH at chr1q, chr6p, chr10q and chr11q.

Then, we designed three simulation scenarios with six settings to mimic the common challenges existing in real-world data, like the samples used above. Specifically, the three scenarios are (1) Sub-clonal division within the tumor populations via allele-specific copy loss, (2) Reference cells downsampling to minor quantities with 5 or 10 cells, and (3) Tumor cells downsampling to limited quantities with 5, 10, or 22 cells. The ground truth CNA profiles of these simulations are shown in Supplementary Fig. 18 and Fig. 6c, and the outputs and performance of XClone, Numbat, or InferCNV are shown in Fig. 6b, d, Supplementary Fig. 19–21, and Supplementary Data 12. Take the clone-specific allele loss as an example (Fig. 6b, c), where the top 3 methods (XClone, Numbat, and inferCNV) were compared. Most remarkably, XClone shows superior performance in identifying copy loss (AUROC: 0.2971 by Numbat, AUROC: 0.9568 by inferCNV VS 0.9964 by XClone, Fig. 6b). Although inferCNV detects copy loss for all tumor cells, it

cannot detect allele-specific copy loss on chr3, hence missing the subclonal structure (Supplementary Fig. 19d). For Numbat, it entirely missed the copy loss in chr3 (for both alleles; Supplementary Fig. 19c), which may result from the initial preprocessing stage that delineates CNA segments using bulk-level data. This approach could focus the analysis on regions containing CNAs in a simple dataset, but it might also result in incorrect identification of areas with intricate clonal CNA structures, such as the two subclones presented in this simulation. On the other hand, XClone, with its higher cellular resolution, has a greater potential to circumvent this issue and accurately resolve complex CNA patterns at the single-cell level (evidenced by both the smoothed BAF signal and final CNA calls in Supplementary Fig. 19a, b). Other than that, Numbat and XClone achieved comparably high accuracy in detecting CN-neutral LOH and copy gain, while Numbat reports many erroneous allele-balanced copy gain partly due to its wrong detection of chr3 as CN-neutral (Supplementary Fig. 19c).

Compiling all simulation results together can further provide a detailed guideline to users for them to select methods when analyzing their datasets. Here, we summarize these three guidelines as follows,

1. When clone-specific allele loss exists (Supplementary Fig. 19), XClone is capable of detecting both alleles, while Numbat does not support this complex scenario and InferCNV can only detect the absolute copy loss ignoring the allele information and potential subclones (this mimics the BCH869 dataset).
2. When the tumor cells are too few, XClone remains capable of working, while Numbat fails to output anything. This may reflect the erroneous results in ATC2 data where the tumor cells are the minority. Of note, inferCNV shows good robustness for low tumor prevenance, if one is mainly concerned about the absolute copy number.
3. When the number of reference cells is as low as 5, both XClone and Numbat can still work well.

## Discussion

In summary, XClone is a statistical method that models CNA states with RDR and BAF modules, and effectively smooths both cell neighborhood and gene coordinate graphs. We applied XClone to four unique cancer samples (glioma, ATC, TNBC, and astrocytoma), each with distinct features such as complex clonal structure, extremely low purity, ubiquitous CNAs, and whole genome duplication. XClone demonstrated its robust and powerful capability in detecting allele-specific CNA states and capturing a broad spectrum of genetic variations, which can help reveal their phenotypic impact.

Moreover, equipped with an accurate BAF module, XClone has demonstrated its power in detecting allele-specific loss (for example in the glioma BCH896) and widespread LOH (for example TNBC1). Such allelic information is critical to reveal biological insights on cancer-related genes via a regulatory role, as evidenced in the elevated proportion of deferentially expressed genes in LOH regions in TNBC1. Additionally, from the astrocytoma sample, XClone showed the capability of suggesting whole genome duplication and increased burden of chromosomal instability. Also, thanks to its high accuracy in detecting CNAs at the cell level, XClone has the potential to enable the discovery of fine-resolved subclonal structures, providing insights for therapeutic strategies.

Furthermore, with the capability of detecting allele-specific CNAs, our method also opens multiple opportunities to broaden the CNA analysis. First, such cell-level CNA states can be further employed to infer the evolutionary processes in a cancer patient, potentially with treatment. This can be even more useful when combined with other types of somatic mutations, e.g., mitochondrial variations[20], especially with enriched protocols like MAESTER[21]. Second, with the rapid development of spatial transcriptomics (ST) technologies, allele-specific CNA detection can further increase the subclonal structure resolution in a spatial context[22], hence allowing us to study the cell-cell

interaction not only between cell types but also between tumor clones[23]. Methodologically, the spatial information may also, in turn, serve as another factor for CNA smoothing, hence stabilizing the CNA analysis in such extremely sparse ST data. Third, the ability to detect CNA in scRNA-seq data further allows us to compile a pan-cancer catalog of common CNAs and their functional impact in a general and context-specific manner.

## Methods

### Overview of XClone suite

XClone offers a comprehensive end-to-end pipeline for CNA analysis using scRNA-seq data. This pipeline takes bam file(s) as its input, models raw UMI or read counts, and outputs both smoothed BAF and RDR matrices for visualization, as well as the detected cell-by-gene CNA states as final results. It also requires users to designate a diploid cell group as a reference. This process comprises four main stages: pre-processing, pre-analysis, modeling, and post-analysis. These stages are thoroughly explained below and illustrated in (Fig. 1a).

### Preprocessing: extract count matrices with xcltk

To facilitate the pre-processing of single-cell data for XClone, we have developed a toolkit called xcltk, which can be found at https://pypi.org/project/xcltk/. This toolkit is designed to extract cell-by-gene count matrices from one or more BAM files, serving both RDR and BAF modules (as illustrated in Fig. 1b). The input can either be a single BAM file with embedded cell barcodes (like those produced by 10x Genomics) or multiple BAM files with each file corresponding to a single cell (such as those generated by Smart-seq2). By default, xcltk uses gene annotations from GENCODE (32,696 genes for hg19 and 33,472 genes for hg38), aligning with the approach taken by Cell Ranger. However, users have the flexibility to use any custom gene (or other feature) annotations.

To produce generate allele-specific counts (upper panel in Fig. 1b), xcltk necessitates a Variant Call Format (VCF) file that contains phased germline genotypes from population haplotype (see next subsection), along with the standard lists of genes and cells. The tool pinpoints all heterozygous variants and calculates UMIs or reads supporting each allele. Consequently, this process generates two matrices that present the alternative allele (AD) and depth (DP) for each SNP within each cell. Making use of genotype phasing information, xcltk can merge multiple SNPs from a single gene and condense both AD and DP into cell-by-gene matrices. It is important to highlight that xcltk avoids double-counting when a UMI or read spans multiple variants within a single gene. This is accomplished by preserving the UMI or read IDs for each SNP and ensuring they are only counted once during the aggregation of all SNPs.

In terms of read depth count generation (lower panel in Fig. 1b), xcltk computes the UMIs or read counts for each gene in every cell, resulting in a sparse matrix. The process is highly comparable to the approach adopted by Cell Ranger, a tool developed by 10x Genomics, which includes filtering of reads (by default, reads with MAPping Quality (MAPQ) < 20, aligned_length < 30nt or Combination of bitwise flags (FLAG) > 255 are filtered out).

### Pre-analysis: three-level allele phasing for haplotype-specific BAF

In XClone, in order to enhance the signal-to-noise ratio when analyzing the allele information in the BAF module, we utilized a three-level allele phasing approach. First, we implemented reference-based phasing (Fig. 1b) by utilizing the human population haplotype reference to phase individual SNPs. This phasing was accomplished by EAGLE2[24] with the Haplotype Reference Consortium panel[25], which was powered by the Sanger imputation server (https://imputation.sanger.ac.uk) as the default option. Once the individual SNPs were phased, xcltk was able to count the phased alleles for SNPs across a gene (as described in Section "Preprocessing: extract count matrices with xcltk").

Second, to avoid the errors in the population-based phasing when it comes to a longer genome range covering many genes, we further performed a gene-bin phasing (local phasing; Fig. 1c) to link a group of proximate and consecutive genes (we call it a gene bin, by default 100 genes). Intuitively, in each cell, we force the allele frequency AF to be the same for all genes in a gene bin, allowing flipping the allele of each gene. Namely, it assumes the CNA states are the same within a gene bin in a cell, even though the CNA states can be different from cell to cell (e.g., some are normal cells and some are tumor cells). Similar to CHISEL[6], we implemented an EM algorithm (Supplementary Algorithm 1) to achieve this by flipping the allele of each gene within a gene bin.

Third, for synchronizing the BAF for visualization, we introduced a chromosome-range phasing (global phasing Fig. 1c) to further link all gene bins on each chromosome arm. Similar to the gene-bin phasing, this step can also use the genomic auto-correlation on the CNA states, which could be addressed by a dynamic programming algorithm, a HMM, or a Gaussian process model. As this step is relatively independent of CNA state identification, we adopted the dynamic programming algorithm proposed in CHISEL for simplicity and efficiency (see Supplementary Algorithm 2). Note, both the gene-bin and chromosome-range phasing steps depend on the auto-correlation of BAF deviance from 0.5 (theoretical BAF value for allele balance). Otherwise, if there is no allelic imbalance, it will return arbitrary phasing, which is irrelevant to the CNA state inference. The three-level allele phasing illustration is shown in Supplementary Fig. 1 and the remarkable performance of the phasing of the three datasets was visualized in Supplementary Figs. 3, 7, and 9, for sample BCH869, ATC2 and TNBC1, respectively.

### Pre-analysis: horizontal and vertical smoothing for visualisation

Before detecting the CNA states, it is beneficial to visualize the raw signal of both RDR and BAF modules. Due to the high technical sparsity in scRNA-seq data, it is necessary to perform smoothing for the visualization. Here, we performed both a horizontal smoothing by a weighted moving average (WMA) along with each chromosome and a vertical smoothing across cells via a KNN graph. Both views of smooths are commonly used in scRNA-seq analysis, for example, WMA is the core smoothing method in InferCNV[26] and the KNN smoothing plays an important role in smoothing unspliced RNAs in scVelo for RNA velocity analysis[27].

Taking the RDR module as an example, the raw count matrix was first normalized to its total library counts, then divided by reference cell expression profile, and finally transformed to the logarithm scale as log ratio. Then this raw RDR matrix $X$ is smoothed horizontally by a WMA connectivity matrix $W$ (via right matrix multiplication) and vertically by a KNN connectivity matrix $M$ (via left matrix multiplication), as follows,

$$\tilde{X} = M \times X \times W, \tag{1}$$

where $M$ is a cell-by-cell connectivity matrix generated from the KNN graph ($k = 10$ by default) of routine scRNA-seq analysis scanpy.pp.neighbors(), and $W$ is a gene-by-gene connectivity matrix linking each gene to its proximate consecutive genes on each chromosome arm (window size $t = 40$ by default), as follows

$$W_{j,g} = \frac{W_{j,g}^o}{\sum_h W_{h,g}^o}, W_{j,g}^o = \begin{cases} t - |d(j;g)|, & 0 < |d(j;g)| < t \\ t * 2, & d(j;g) = 0 \\ 0, & |d(j;g)| \geq t, \end{cases}$$

where $d(j;g)$ refers to the ordinal distance from any gene $j$ to a query gene $g$ on genome coordinate. Of note, $W$ is column normalized and $M$ is row normalized, namely with sum to one. Also, both matrices have higher weights for each node itself, hence balancing the denoising and signal preserving.

For the B allele frequency (BAF) module, we took the cell-by-gene_bin matrix as input and performed a similar smoothing both horizontally (with WMA, window size $t = 101$ by default) and vertically (with KNN). As an example, the smoothing steps for BCH869 scRNA-seq were shown in Supplementary Fig. 3. Once the smoothed RDR and BAF matrices are generated, they can serve both visual representation and empirical extraction of CNA parameters (see Sections "Modeling: mixture model for expression data (RDR module)" and "Modeling: mixture model for phased allele-specific data (BAF module" below)).

## Modeling: mixture model for expression data (RDR module)

The RDR module aims to detect the states of absolute copy numbers, i.e., CN-neutral, loss, and gain, for each gene in each cell. To start, it requires specifying reference cells (i.e., diploid cells), which can be picked from pre-annotated cell types from transcriptome by examining markers of non-tumor cells. Given the reference cells, XClone will calculate the reference average gene expression (ref_avg) and filter the genes with low expression (default 0.5 for 10x genomics and 1.8 for smart-seq). Optionally, it will further remove top marker genes for each pre-annotation cell group (15 by default), e.g., as detected by the scanpy.pp.rank_genes_groups().

Then, we used a negative binomial distribution to model the observed UMI (or read) counts for each CNA state in a Generalized Linear Model (GLM) framework, as follows

$$p(X_{c,g}|z_{c,g,k}=1) = \text{NB}(X_{c,g}|\mu_{c,g,k},\phi_g)$$
$$\mu_{c,g,k} = l_c \times X_{ref,g} \times C_k, \qquad (2)$$

where $X_{c,g}$ denotes the raw read/UMI count at gene $g$ cell $c$; $l_c$ denotes the cell-specific library size factor; $X_{ref,g}$ denotes the average count value for gene $g$ in given reference diploid cells; $C_k$ denotes the copy number ratio compared to reference diploid state for each CNA state. The latent variable $z_{c,g,k}$ indicates if gene $g$ in cell $c$ belongs to CNA state $k$ ($z_{c,g,k}=1$) or not ($z_{c,g,k}=0$). In this module, we defined three states with index $k=0$ for copy loss, $k=1$ for copy neutral, and $k=2$ for copy gain.

Considering that the reference count values $X_{ref,g}$ can be obtained directly, it leaves only three sets of parameters to be estimated: $l_c$, $C_k$, and $\phi_g$. Here, we estimated them separately by a maximum likelihood estimation with the statsmodels package[28] or using a heuristic method. To estimate the library size factor $l_c$ for each cell $c$, by default, we used the ratio of its total sum count to the average of reference cells. To estimate the gene-specific dispersions $\phi_g$, we fit a negative binomial GLM per gene in the form of Eq. (2) but only on the reference cells ($C_k=1$) with the fitted library size $l_c$ from the previous step.

For the CNA ratio $C_k$, one can directly use a theoretical value, namely {0.5, 1, 1.5}, respectively, for loss, neutral, and gain (Supplementary Fig. 2). Here, we further employed a heuristic method from the empirical distribution. Based on the smoothed RDR value, we selected 3 guided genomic regions (chromosomes or arms) that are most likely to be copy loss, copy neutral, and copy gain regions, as ranked by the mean. In general, CNA regions only cover a small percentage of the whole cell-by-gene matrix and the extreme ratio value at each CNA state can be used to guide CNA ratio initialization. Thus, the CNA ratio $C_k$ will be extracted from quantiles of each of the corresponding chromosomes (or arms), specifically default quantile 0.96 in copy neutral, quantile 0.0001 in copy loss, and quantile 0.99 in copy gain regions.

Once all three sets of parameters are fitted via the above procedures, we can calculate the probability of assigning each gene in each cell to a certain CNA state via Eq. (2). Namely, we can obtain a cell-gene-

state probability tensor, as follows

$$p(z_{c,g,k}=1|X_{c,g},l_c,X_{ref,g},\boldsymbol{C}) = \frac{\text{NB}(X_{c,g}|l_c \cdot X_{ref,g} \cdot C_k,\phi_g)}{\sum_k \text{NB}(X_{c,g}|l_c \cdot X_{ref,g} \cdot C_k,\phi_g)}, \qquad (3)$$

where NB() is the probability density function of negative binomial distribution parameterized by the mean and dispersion. Once the CNA state assignment probability is calculated above, we can have a hard assignment of each state by taking the state with the highest probability. Of note, this probability will be further smoothed cross cells by a kNN smoothing and cross genes by an HMM smoothing (see Section "Post-analysis: CNA states smoothing, combination and denoise" below).

## Modeling: mixture model for phased allele-specific data (BAF module)

For the BAF module, we took both AD and DP sparse matrices from xcltk, removed marker genes provided by the RDR module, and performed the three-level phasing to generate gene_bin based matrices as inputs for XClone modeling. Here, we used a beta-binomial distribution to model the observed UMI (or read) counts for each CNA state, as follows

$$p(a_{c,g}|d_{c,g},z_{c,g,k}=1) = \text{BB}(a_{c,g}|d_{c,g},\rho_{g,k},\tau_{g,k}), \qquad (4)$$

where $a_{c,g}$ and $d_{c,g}$ denote phased B allele count and total count at gene_bin $g$ in cell $c$, respectively. The Beta-binomial is parameterised by the mean success rate $\rho$ and concentration $\tau$ (namely $\alpha = \rho\tau$ and $\beta = (1-\rho)\tau$ in conventional parameterization). Here, we treat the concentration as a technical factor and shared by all gene_bins in droplet-based data ($\tau_{g,k}=100$ by default) while it can vary in smart-seq data depending on the coverage of each gene_bin.

Here, $z_{c,g,k}$ is a latent variable indicating the cell $c$ in gene_bin $g$ is CNA state $k$ ($z_{c,g,k}=1$) or not ($z_{c,g,k}=0$). The parameter $\rho_{g,k}$ denotes the expected allele frequency of gene_bin $g$ in CNA state $k$. The state index $k = \{0, 1, 2, 3, 4\}$ is respectively for allele A strong bias, allele A minor bias, allele balance, allele B minor bias, and allele B strong bias, with the corresponding theoretical allele frequencies {0, 1/3, 0.5, 2/3, 1} (Supplementary Fig. 2).

The dataset-specific allele frequency parameters $\rho_{g,k}$ can also be estimated from the horizontally smoothed BAF matrix to capture more accurate CNA patterns. Specifically, we first estimated the allele frequency for copy neutral state $\rho_{g,k=2}$ from reference cells by directly using its gene_bin-specific mean allele frequency. If there are too few reference cells to accurately estimate the allele frequency, it can be set to 0.5 by default. Next, to estimate the allele frequency of other BAF states $k = \{0, 1, 3, 4\}$, we applied a Gaussian mixture model to the smoothed BAF matrix to identify five components with mean values, in order, for allele A bias (strong, minor), allele balance and allele B bias (minor, strong) as an empirical estimation of the CNA ratio for the BAF model. XClone also provides another mode of 3 BAF states (allele A bias, allele balance, and allele B bias) with the corresponding theoretical allele frequencies {0, 0.5, 1} visualization for comparison.

When the parameters are optimized above, similar to the RDR module, we can also obtain a cell-by-gene_bin-by-state likelihood tensor by normalizing Eq. (4), as follows,

$$p(z_{c,g,k}=1|a_{c,g},d_{c,g},\rho_{g,\cdot},\tau_{g,\cdot}) = \frac{\text{BB}(a_{c,g}|d_{c,g},\rho_{g,k},\tau_{g,k})}{\sum_k \text{BB}(a_{c,g}|d_{c,g},\rho_{g,k},\tau_{g,k})}, \qquad (5)$$

where BB() is the probability density function of beta-binomial distribution parameterized by the success rate and concentration. Then, similar to the RDR module above, we can also obtain the hard assignment of the BAF-based CNA state by taking the one with the highest probability, after it gets smoothed.

**Post-analysis: CNA states smoothing, combination and denoise**

Instead of directly using Eqs. (3) and (5) for CNA states assignment at each gene (or gene_bin) in each cell respectively in RDR and BAF modules, we further introduced a smoothing strategy for the state's assignment. Specifically, for each module separately, we first performed a vertical smoothing via a KNN connectivity matrix, as used in Section "Pre-analysis: horizontal and vertical smoothing for visualisation", i.e., via the left matrix multiplication with $M$ in Eq. (1). Then we treated it as the input emission probability matrix for horizontal smoothing by a HMM. For HMM, the start probability and transition probability can be customized to capture different resolutions of smoothing. Here, we set start probability {0.1, 0.8, 0.1} (10x Genomics), {0.3, 0.4, 0.3} (smart-seq) for the RDR module and {0.2, 0.15, 0.3, 0.15, 0.2} for the BAF module, and transition probability ($\{t = 10^{-6}, 1-(K-1)t\}$ respectively for cross-state transition and state keeping, Supplementary Fig. 1g) as default. Then, we employed a forward-backward algorithm in HMM to achieve a maximum likelihood for CNA assignment probabilities along genomic coordinates (see more details about this algorithm in Chapter 13 in ref. 29).

To combine the CNA states of RDR and BAF modules, we first need to synchronize feature dimensions, as XClone's RDR module outputs a cell-by-gene-by-state CNA posterior probability tensor, while BAF has a shape of cell-by-gene_bin-by-state. The default way is to map the BAF gene_bin scale back to the gene scale according to the RDR module. Then, we can combine the same dimensional RDR and BAF CNA posterior probability tensors following the combination strategy in Supplementary Fig. 2 to get the final combined CNA posterior probability tensor.

After XClone detects the CNA states for each gene in each cell, a post-denoise step can be optionally applied to clean the regions with a low chance of carrying CNAs. Specifically, for each gene, we calculated the proportion of aneuploid cells and then fit a two-component Gaussian mixture model over the aneuploid cell proportion among all genes. Then the genes assigned to the bottom component (i.e., with the lowest proportion of aneuploid cells) will be taken as neural CNA regions, which hence will be masked for CNA analysis. Alternatively, users can manually set a threshold as cell proportion to denoise the probability matrix if there is other evidence. We apply the post-denoise strategy default on the BAF module and it also can be applied on the RDR and combined module by manually setting it.

**Data generation, download, preprocessing and analysis**

The scRNA-seq data for a fresh gastric cancer tissue sample GX109-T1c was generated by 10x Genomics Chromium 5′-scRNA-seq platform at the Centre for PanorOmic Sciences (CPOS) at HKU. Sample was collected with patient's informed consent. The study was approved by the Institutional Review Board of the University of Hong Kong and the Hospital Authority Hong Kong West Cluster (IRB reference ID: UW14-257) and complies with all relevant ethical regulations, including written informed consent from the patient for data release. Sufficient details of the experiments are provided in Supplementary Methods.

The raw FASTQ files were aligned to the human genome (GRCh38) via Cell Ranger v2.2.0. In total, 5400 cells were called with the same pipeline described in[30,31] (Details in Supplementary Methods) and were analyzed with Seurat by selecting the top 2000 highly variable genes and using 50 top principal components, followed by Leiden clustering with resolution 0.1, resulting in 7 clusters (Supplementary Fig. 17). We further manually annotated these clusters according to the cell type marker genes in Supplementary Table S2 from[32] and aggregated them into 4 major different cell types: endothelium/fibroblast (endothelium or fibroblast cell), epithelium, immune cells, and stem/cancer cell (gastric stem or cancer cell), respectively with cell counts of 63, 20, 3164 and 2153.

The BCH869 data generated using smart-seq2 was downloaded in FASTQ files from GEO (GSE102130) by request from the authors.

Within the original BCH869 dataset, there are 489 cells, with four tumor subclones composed of 227, 221, 34, and 7 cells, according to the original study[14], 3 normal cells (Oligodendrocytes and immune cells) from the same sample were combined in the dataset as reference cells in our study.

ATC2 is an anaplastic thyroid cancer sample with 6224 cells where tumor purity was found extremely low[9]. TNBC1 is a triple-negative breast cancer sample with 1097 cells (2 subclones in tumor cells)[9]. The ATC2 and TNBC1 datasets were generated using 3′ 10x Genomics platform by Gao et al.[9] and the aligned reads in BAM format and the called cell list were directly downloaded from GEO (GSE148673). The GBM is a second recurrence (2R) IDH1-mutant astrocytoma sample with 4416 cells sequenced by droplet-based snRNA-seq in ref. 18. For more details on the 5 datasets, see Supplementary Data 1.

Given the aligned BAM file(s) and cell lists, we performed the genotyping by using cellsnp-lite v1.2.0[13] and genotype phasing by using the Haplotype Reference Consortium panel[25] with EAGLE2[24], powered by Sanger Imputation Server (https://imputation.sanger.ac.uk). Then, feature counting was achieved by using xcltk v0.1.15 (see above Section "Preprocessing: extract count matrices with xcltk").

**Benchmarking CNA detection performance**

It is necessary to benchmark the CNA inference methods using gold standard annotation as ground truth. The CNA profile of the BCH869 sample was well characterized by WGS and scRNA-seq, which is used as ground truth to identify copy number loss, copy number gain, and loss of heterozygosity. The BCH869 dataset has a complex clonal annotation (4 subclones with different CNA events) where the clone-specific CNA annotation was determined from the combination of scRNA-seq data (by InferCNV) and bulk WGS with haplotype analysis (BCH869 CNA ground truth in Supplementary Data 2, Number of consensus cells and genes of five methods for benchmarking on BCH869 in Supplementary Data 3).

With ground truth of BCH869 sample, we benchmarked XClone with four other CNAs detection tools specifically designed for scRNA-seq data, InferCNV[26], CaSpER[11], CopyKAT[9] and Numbat[12] (XClone v.0.3.4, InferCNV v.1.8.0, CopyKAT v.1.0.4, CaSpER v.0.2.0 and Numbat v.1.2.1) using their default parameters (details in Supplementary Methods and running efficiency in Supplementary Data 4).

For CNA detection performance evaluation at the single-cell level, we built a consensus cell-by-gene signal matrix for each tool at each CNA state. For XClone and Numbat, CNA event posterior probability was used to build a cell-by-gene matrix, and for InferCNV, CaSpER, and CopyKAT, the modified expression was used. For Numbat and CaSpER whose output is at genomic region resolution, we mapped them to the gene level. The ground truth matrix is a same-dimensional cell-by-gene binary matrix indicating the existence of CNA events in the consensus cells and genes obtained above. The ground truth matrices are prepared for each state (copy gain, copy loss, and LOH) independently. Ground truth matrix value 1 indicates the CNA exists and 0 means not. By flattening the matrix and varying the cutoff on signal scores, we calculated the true positive rate (TPR) and false positive rate (FPR) between the detected (cell-by-gene signal matrix) and real CNA regions (cell-by-gene binary ground truth matrix) for each tool, further generated a ROC curve and calculated the area under the ROC curve (AUC) values for evaluation. It is worth mentioning that for fair benchmarking with different tools, XClone can output CNAs in two resolutions: (1) gene_scale, where the CNA probabilities are reported for each gene in each cell; (2) chromosome arm_scale, where the CNA probabilities are reported for each arm of chromosomes in each cell. We showed the gene scale benchmarking results in Fig. 2f–h and the chromosome arm scale benchmarking results in Supplementary Fig. 5 for BCH869. Additionally, we also showed the Precision-Recall curves of each method at both the gene scale and arm scale for BCH869 in Supplementary Fig. 6.

Further, we take the ATC2 sample with extremely low tumor purity to benchmark the subclone identification performance. We found that inferCNV and CopyKAT both detect 2 subclones obviously (Fig. 3g, h). By Combining expression with BAF analysis, we annotated the non-normal cells clusters into subclone1, and subclone2, which we used as cell annotation for assessing the subclonal detection performance for XClone by calculating the Adjusted Rand Index (ARI) metric (0.998 compared with inferCNV; 0.961 compared with CopyKAT, and 0.963 for inferCNV with CopyKAT, Fig. 3a, bottom panel).

For the TNBC1 sample, as a high-quality annotation is lacking and it implies complex CNA profiles, we only used it in a qualitative manner by examining the consistency between the smoothed BAF signal (without inference) and the inferred CNA states. In this sample, the original paper reports two major subclones that comprised 44% (subclone A) and 28% (subclone B) of the tumor mass within the identified tumor cells[9]. By Combining expression with BAF analysis, we annotated the cell clusters into Clone 1, Clone 2, and Normal cells, which we used as cell annotation for assessing the clonal reconstruction performance for XClone by calculating the ARI metric (0.9653; Fig. 4e).

Then, we add a second recurrence astrocytoma sample with a newly emerged minor clone to perform the benchmarking. This GBM sample has undergone snRNA-seq from 10x Genomics, bulk whole exome sequencing (WES), and scOne-seq, which concurrently profiles DNA and RNA within individual cells with exceptional efficiency. Consequently, this sample serves as an excellent dataset with CNA ground truth for benchmarking. The CNA ground truth file of the minor clone (2R clone 1, with 53 cells) is in Supplementary Data 7. The number of consensus cells and genes used in the benchmarking are listed in Supplementary Data 8.

These evaluations provide an understanding of the strengths and weaknesses of each CNA detection method, highlighting the areas where improvements might be needed. The insights gained from this comparison can guide the choice of methods in future studies, depending on the specific features of the CNAs that are of interest.

### Differential gene expression analysis

We conducted a differential gene expression analysis on the BCH869 glioma dataset, by utilizing the "rank_genes_groups" function in "scanpy". This analysis aimed to identify statistically significant differences (absolute log2fc > 1 and adjusted $p < 0.05$) in gene expression between clone 1 and clones 2–4 on chromosome 14, where allele-specific loss is observed.

The same Scanpy function was used to perform differential expression analysis on the TNBC1 breast cancer dataset. Comparisons were made between Clone 1 and Normal, and Clone 2 and Normal to determine DEGs. Subsequently, the "enrichGO" function in "ClusterProfiler"[33] was used to perform gene set enrichment analysis on the DEGs, thereby identifying biological processes that are over-represented in these genes.

Both the differential expression analysis and gene set enrichment processes were adjusted for multiple testing through False Discovery Rate (FDR) estimation, using the Benjamini–Hochberg (BH) method for $p$-value correction.

### Seed dataset GX109-T1c and the generation of BAM files for different simulations of challenging scenarios

Our scCNAsimulator requires a reliable seed dataset as input to generate sequencing reads (supporting UMIs in a BAM format) over allelic CNA profiles (See the details in Supplementary Technical Notes). To support this, we used a clean dataset generated by an in-house collected fresh gastric cancer tissue GX109-T1c. This tissue was processed to yield single-cell RNA sequencing data, encompassing 5400 cells using the 5′ end protocol from 10x Genomics, as well as single-cell DNA sequencing data obtained through the 10x Genomics. Additionally, a bulk WES was conducted on a frozen sample of the same tissue

(GX109-TF, sampled from an area contiguous with the blocks used for scRNA-seq and scDNA-seq, Supplementary Fig. 15) to provide a complete genomic picture. The CNA states and regions called from both bulk WES (via CNV-facet tool[34]) and scDNA-seq (via 10X CNV kit and CHISEL[6]) are highly concordant, including copy gain at chr8, chr20, copy loss at chr12p (chr12:7288360-16757762) and chr18, and CN-neutral LOH at chr1q, chr6p, chr10q and chr11q (Supplementary Figs. 15, 16; Supplementary Data 1 and 9). Additionally, the scDNA-seq CNA clonal structure is relatively simple in this sample, consisting of two major subgroups, with/without copy number changes. Following the regular analysis pipeline[30], the scRNA data clustering of this sample further confirmed this clonal structure by separating the cells into two significant populations: one (3227 in 5400, 59.8%) was classified as tumor microenvironment cells with immune or fibroblast markers expressed, the other one (2153 in 5400, 39.9%) was classified as gastric stem/carcinoma cells with strong expression of stem cell markers and EPCAM (Supplementary Fig. 17). These two scRNA groups are considered concordant with the two scDNA-seq groups, respectively, with CN neutral and harboring CNAs, hence it can serve as an ideal seed dataset for our simulator scCNASimulator to generate synthetic data in diverse scenarios, as listed below.

**Simulation 1: Sub-clonal division within the tumor population.** For the 2153 cells categorized as "stem, cancer cell" we created two sub-populations of 1076 and 1077 cells. Each sub-group exhibited a unique allele copy number loss on chromosome 3 (See the details in Supp technical notes). We achieved this by 3 main steps: (1) Sub-clone Assignment: Assign each tumor cell to one of the sub-clones. Split the 2153 tumor cells into two groups of approximately equal size (1076 and 1077 cells). (2) Phasing: Use the output from Sanger Phasing, which provides allele-specific UMIs for chromosome 3 in each tumor cell. This phasing information assigns UMIs to specific alleles, allowing for the simulation of allele-specific copy number loss. (3) Modify the BAM File: Allele-Specific UMI Discarding or Retention: For each sub-clone, it discarded allele-specific UMIs from the BAM file to simulate a copy number loss for that particular allele on chromosome 3. This means removing UMIs that correspond to the allele with the copy number loss in one sub-clone but retaining them in the other. Discard Ambiguous UMIs: Remove any UMIs that do not have clear allele information ("ambiguous" UMIs) at the probability of 0.5 (mimic the natural state) to avoid introducing noise or confounding factors in the CNA analysis.

**Simulation 2: Reference cell downsampling to minor quantities.** The downsampling process for reference cells effectively reduced the "immune cells" population in the BAM file to smaller, more specific subsets. Here is a step-by-step explanation of how the downsampling was performed: (1) Selection of Cell Barcodes: From the total pool of 3164 "immune cells", we randomly selected a predetermined number of cell barcodes corresponding to the subset sizes we wanted to analyze 10, or 5 cells. (2) Filtering of UMIs: Using the selected cell barcodes, we then filtered the BAM file. This involved removing all UMIs that were associated with cell barcodes not included in the selected subsets. This ensures that the BAM file only contains data from the 10 or 5 "immune cells" that we aim to analyze. This simulation process was supported by samtools.

```
$ samtools view -h in.bam | grep -F -f selected_barco-
des.txt | samtools view -b > out.bam
```

In this pseudocode, `selected_barcodes.txt` would be a text file containing the list of sampled cell barcodes that we want to keep.

**Simulation 3: Tumor cell downsampling to limited quantities.** Similar to the simulation in reference cells, from the 2153 "stem, cancer cell" group, we sampled smaller cohorts of 22, 10, and 5 cells to generate 3 different simulated BAM files.

All of the tasks can be achieved by our customized tool scCNASimulator.

## Reporting summary

Further information on research design is available in the Nature Portfolio Reporting Summary linked to this article.

## Data availability

The BCH869 scRNA-seq data (Smart-seq2) raw FASTQ files were downloaded from GEO (GSE102130). The ATC2 scRNA-seq data (3′ 10x Genomics) BAM files were downloaded from GEO (GSM4476492). The TNBC1 scRNA-seq data (3′ 10x Genomics) BAM files were downloaded from GEO (GSM4476486). The GBM snRNA-seq data (10x snRNA-seq data) BAM files were requested from the authors (GSE185269). To perform the simulation, we generated a high-quality seed dataset on a gastric cancer sample GX109-T1c where the sequencing data has been deposited on GEO under accession code (GSE232733) and ENA (PRJEB60922). The raw whole exome-seq data of GX109 tumor tissue and blood control (EGAD00001015382) and its scDNA-seq data (EGAD00001015383) have been deposited on EGA under accession code (EGAS00001007854). The summary of WES data, as the CNA ground truth generated in this study, is provided in the Supplementary Data file (Supplementary Data 9). The processed GX109-T1c scRNA-seq data is available at GitHub https://github.com/Rongtinting/xclone-data. All other data supporting the findings described in this manuscript are available in the article and its Supplementary Information files.

## Code availability

XClone[35] is implemented as an open-source Python package, publicly available at https://github.com/single-cell-genetics/XClone. XClone documentation and tutorials are publicly available at https://xclone-cnv.readthedocs.io/en/latest/. The preprocessing toolkit xcltk is freely available at https://pypi.org/project/xcltk/. The scripts for benchmarking can be found at https://github.com/Rongtinting/CNV_calling_Benchmark. The simulator developed for the allelic loss simulation is available at https://github.com/hxj5/scCNASimulator with detailed application notes.

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

## Acknowledgements

We thank Prof. Angela Wu, Prof. Jiguang Wang, Dr. Lei Yu, and Dr. Quanhua Mu for the scOne-seq data (GBM) sharing and discussion, and the Centre for PanorOmic Sciences (CPOS) at HKU for providing single-cell genomic analysis and flow cytometry services. This work was partially supported by a donation from the Hong Kong Jockey Club Charities Trust, a theme-based research grant from the Research Grants Council of the Hong Kong Special Administrative Region, China (Project No. T12-710/16-R, S.Y.L.), Centre for Oncology and Immunology and Centre for Translational Stem Cell Biology under the Health@InnoHK Initiative funded by the Innovation and Technology Commission, The Government of Hong Kong SAR, China, NSFC (Project No. 62222217, Y.H.) and the University of Hong Kong (a startup fund and a seed fund, Y.H. and a University Postgraduate Fellowship, R.H.).

## Author contributions

Y.H. and O.S. conceived the study. R.H. and Y.H. designed the statistical model. R.H. led the implementation of the XClone software. R.H. performed all data analysis with help from X.H. X.H. implemented the xcltk preprocessing pipeline. X.H. and R.H. performed the benchmarking and simulation part. Y.T. performed the cell annotation for GX109-T1c scRNA-seq data. H.H.Y. and S.Y.L. contributed to the GX109-T1c data and biological insights. S.Y.L. contributed to funding acquisition. R.H. and Y.H. wrote the manuscript. All authors provided feedback on and approved the paper.

## Competing interests

The authors declare no competing interests.
