## [Peer Review File · Nature Communications]

Robust analysis of allele-specific copy number alterations from scRNA-seq data with XCloneReviewers' comments:

Reviewer #1 (Remarks to the Author):

Huang et al, reported Xclone, a tool to analyze allele-specific copy numbers from scRNAseq data. The tool is potential very useful as it improves CNV detection by adding phasing information. However, the resolution might be too low (maybe chromosomal arm levels). CNV segmentation was not pursued. There are concerns regarding the uniqueness, comparing a very similar existing tool, Numbat and the fairness of benchmarking comparisons. The figures are sloppy, and writings need to be improved.

1. First of all, it's strange to write the Xclone algorithm in the introduction section. The tool needs to be described in detail in the Result section. Algorithmically, how does Xclone compare to existing tools, such as strength and weakness, difference, and common features. What are the input and output?
2. What is the resolution of XClone? Is it chromosomal arm level?
3. There isn't description of CNV segmentation. Is the smoothing step equivalent to segmentation?
4. It is unclear how HMM and KNN are performed. KNN is not even shown in the overview Figures (Fig1 and Fig S1). It's probably circular logic to use KNN of single cells to smooth data, if it has the hidden assumption that single cell CNVs are similar if their transcriptome profiles are similar, whereas CNV data are often used to investigate the transcriptional differences between difference CNV clones. WMA smoothing appeared in Figures, which is not described either.
5. How CNV states are determined in XClone, as well as in other tools that are benchmarked? The strategies need to be described in detail in addition of Fig. S2. The authors should not expect reader to understand the algorithm by throwing out a Figure panel. What are the definitions of copy loss A and copy loss B. It's confusing to use red to illustrate copy gain, and pink to show copy loss.
6. Is Fig.1f and Fig.S4a the same? Both are Numbat results of BCH869 data according to the legends.
7. Figure panels and labels need to be improved. Many fonts are too small to read.
8. How ROC values are calculated to compare between different tools? What are the ground truths?
9. Fig.1f, what's the evidence of stating Numbat had a flow in missing copy loss on Chr14 (Line 89), instead of false discovery of Xclone in Fig.1e? Again, how are copy loss A and B decided? Numbat indeed detected the deletion.
10. This is confusing again. Is Fig. S5 same as Fig.1g, h, i? The benchmarking at chromosomal arm scales seems unfair to other tools that may have higher resolution. Is same genomic reference version used to run all tools?
11. In ATC2 data, the major subclonal CNVs in Chr5 are detected in other tools, which is missed by XClone. As such, more datasets are suggested to test Xclone performance. The authors showed 5 panels of XClone results (Fig 2b-e and others). Which is the final result of XClone?
12. What's the difference between Fig. 3c and Fig. S9, the last panel? Fig 3f and Fig. S10a?
13. XClone detected CNVs in normal cells, are they real or false calling? I'd suggest to test Xclone on scRNAseq data of some diploid tumors and normal organs to test the false calling and sensitivity.
14. What's the main advantages of XClone comparing to Numbat, in terms of CNV clone discovery? It seems all tools except for CasPER are able to define clones.

15. Beyond the benchmarking results, what is the unique algorithmic strength/features of XClone comparing to Numbat?

16. How about the technical performance of XClone?

Reviewer #2 (Remarks to the Author):

XClone adds a valuable resource for allele-specific CNV analysis, consistently outperforming its peers in rigorous comparative assessments. My concern revolves around its three operational modes (RAF, RDR, and combined): which one would be the optimal choice for different scenarios, and whether the tool can automatically pick the right one based on users' needs? The manuscript references "Supp. Table S5," which should provide a comprehensive analysis; however, upon review of the Supplementary Materials, this table appears to be missing.

Reviewer #3 (Remarks to the Author):

This manuscript describes XClone, a novel method designed to enhance the identification of allele-specific Copy Number Alterations (CNAs) from single-cell RNA sequencing (scRNA-seq) data when compared to other recent methods. Given the abundance of scRNA-seq datasets in cancer research and the complexity of the problem, any advancements in CNA detection are relevant for cancer sequencing studies. However, this study fails to convincingly demonstrate that XClone adds a substantially novel methodological contribution surpassing previous methods or translates these improvements into the discovery of significantly novel biological insights.

Major Concerns:

1. Lack of Novelty: While XClone does seem to offer some technical improvements over existing methods for identifying allele-specific CNAs from scRNA-seq data, it's important to acknowledge that it is not the pioneering method in this domain. Previous methods, such as Numbat, have pursued similar goals. It is somewhat unclear what novel concepts or features underpin XClone and how these innovations are expected in principle to translate into superior performance compared to Numbat or other similar previous methods. For instance, the principal novel feature of XClone, involving the computation of the BAF signal, appears to be an adaptation of the previous CHISEL method, originally developed for addressing a similar issue in scDNA-seq data. Consequently, XClone seems more like an incremental enhancement of previous methods rather than a substantial novel approach. Consideration of a more technical or algorithmic audience may be apt for this contribution.

2. Insufficient Benchmarking: The absence of a formal benchmark employing a well-established ground truth prevents the demonstration of XClone's improved performance. The provided benchmark relies on a dataset featuring both scRNA-seq and whole-genome sequencing (WGS) data, but it does not establish an authentic ground truth; rather, it relies on inferred results from other methods. Additionally, WGS is not a suitable benchmark for single-cell results, as it is a bulk approach. To validate the method, a robust benchmark with a genuine ground truth is essential. I would recommend the authors to consider using either: (a) existing simulations, as done in [<https://doi.org/10.1101/2023.04.01.535197>] which used scRNA-seq simulator SPARSim [<https://doi.org/10.1093/bioinformatics/btz752>] and added simulated CNAs; or (b) using existing datasets having both scDNA-seq and scRNA-seq data (for example [<https://doi.org/10.1093/nargab/lqaa016>]).

3. Absence of Novel Biological Insights: When XClone is applied to existing datasets, it appears to

produce results largely akin to those generated by at least one of the other existing methods. It does not seem to yield novel biological insights, and the different results obtained between different methods largely appear relatively minor. This also raises concerns about which inference might align more closely with the truth, especially in the absence of orthogonal analyses.

4. Clarity of Text: The manuscript offers limited explanations and is heavily laden with technical terminology, making it challenging to follow, particularly for a non-expert audience. A dedicated Results section that elucidates the novel approach and its unique attributes is also missing, hindering a clear understanding of the anticipated advantages over previous methods and which elements are genuinely innovative versus adaptations from prior approaches. Lastly, the manuscript's brevity leaves room for additional details and explanations. Furthermore, certain methodological sections lack proper mathematical descriptions, further complicating comprehension. For example, one of the main steps of XClone in Section 4.3 lacks any equation or formal mathematical description. Also what are the details of the defined HMM (covariates, model, etc.)? And several other important details are missing.

Minor Terminology Comment:

The manuscript utilizes the term "Copy Number Variants (CNVs)" incorrectly in this context. CNVs are typically associated with germline variants. For somatic alterations in cancer cells, "Copy Number Aberrations/Alterations (CNAs)" is the standard and appropriate terminology.

Point-by-point response

Reviewer 1: pages 2 to 11;

Reviewer 2: pages 12 to 13;

Reviewer 3: pages 14 to 17.

The responses are highlighted in blue.

Reviewer #1 (Remarks to the Author):

Huang et al, reported XClone, a tool to analyze allele-specific copy numbers from scRNAseq data. The tool is potential very useful as it improves CNV detection by adding phasing information. However, the resolution might be too low (maybe chromosomal arm levels). CNV segmentation was not pursued. There are concerns regarding the uniqueness, comparing a very similar existing tool, Numbat and the fairness of benchmarking comparisons. The figures are sloppy, and writings need to be improved.

Response: Thank you for recognizing the value of XClone on single-cell CNV detection with additional phasing information. Indeed, the BAF with the 3-step phasing strategy adds major complementary information and improves the performance of CNV detection.

With your suggestion on the manuscript presentation, we have extended our original concise description with substantially more details in both texts (e.g., new section 2.1, High-level description of XClone model and workflow) and figures (e.g., updated Fig. 1). Please also see the response to points 2 & 3 for the CNV resolution where we clarified the misunderstandings from the reviewer on the CNV resolution, points 8 & 9 for the fairness of benchmarking comparison where you can see we have already employed strategies with high fairness, and points 14 & 15 for the uniqueness of our method where three major novelties have been highlighted that ensure the robustness and accuracy of our method.

In terms of the resolution, there is a fatal misunderstanding. In XClone, the CNV resolution is not at the chromosomal arm level (only for fairly benchmarking with other methods), but actually at the gene level, in each individual cell. We replaced the conventional hard segmentation of CNV with this new strategy of gene-level CNV states that are smoothed with a Hidden Markov Model (HMM). Additionally, all the CNV detection is performed at single cell level (with neighbor cell smoothing), instead of using pseudo-clone as used in Numbat or predefined clones in InferCNV. We expect such settings will better balance the CNV signal detection and the downstream analyses of the cell transcription phenotypes.

For the uniqueness of XClone, we have added a new Result section to fully introduce our method, particularly highlighting its high resolution at gene and cell level, two-view graph-based smoothing and substantially optimized preprocessing. We give more details in the following point-by-point response to elaborate the fairness and comprehension in benchmarking. We apologize if the figures and writing have led to any confusion, and we will ensure to improve the writing in the revised version of our manuscript. Thank you for bringing this to our attention.

1. First of all, it's strange to write the XClone algorithm in the introduction section. The tool needs to be described in detail in the Result section. Algorithmically, how does XClone compare to existing tools, such as strength and weakness, difference, and common features. What are the

input and output?

Response: Admittedly, the manuscript was adapted from a brief communication format to an article, therefore, you may see it is written in a concise style. Now, we have added a new Results section 2.1 (in p.2) to fully describe the uniqueness, strength, weakness and features of our methods, together with the input and output. Thanks for your suggestion.

2. What is the resolution of XClone? Is it chromosomal arm level?

Response: Related to point 3 below, we employed the HMM for a smoothed CNV state assignment, which serves as an alternative way to the CNV segmentation but preserves more original signal. This setting is also the core part of the popular InferCNV and CHISEL, which can ensure higher robustness. Therefore, XClone has a gene (or bin, 100 genes as default) scale resolution and only used chromosomal arm level output for fairer benchmarking to other tools.

Specifically, in the RDR module, the resolution of XClone is gene scale. In the BAF module, through the strategy of 3-step phasing, we obtain more effective BAF information at the bin level (100 genes as default). For the combination of RDR and BAF modules, we map the bin scale BAF to the gene scale and combine RDR and BAF at the gene scale. The final detected CNV is at the resolution of the gene scale. The details are shown in Figure 1a, and we have added more details on p.2 to improve its clarity.

However, the resolution of each benchmarked method is not the same. To fairly compare different methods, we need to choose a unified scale for comparison. InferCNV and CopyKAT detect CNV on a gene scale, while CaSpER and Numbat detect CNV on a near-arm scale.

For fair benchmarking, we added an option for XClone to output chromosomal arm level CNV by merging genes to arm scale and compared the results at different resolutions with all methods (gene resolution in original Figure 1 (g-i), updated Figure 2 (f-h) and chromosomal arm resolution in Figure S5).

3. There isn't description of CNV segmentation. Is the smoothing step equivalent to segmentation?

Response: Yes, the HMM smoothing framework in both the RDR and BAF modules plays a role in segmenting to some extent, therefore we do not need to perform additional segmentation. As mentioned in Point 2 above, this smoothing strategy (together with an orthogonal smoothing across cells via a KNN-based graph) is essential for gene and cell resolution. A similar philosophy of preserving high resolution is also used in CHISEL and InferCNV, which are important to achieve high robustness.

4. It is unclear how HMM and KNN are performed. KNN is not even shown in the overview Figures (Fig1 and Fig S1). It's probably circular logic to use KNN of single cells to smooth data, if it has the hidden assumption that single cell CNVs are similar if their transcriptome profiles are similar, whereas CNV data are often used to investigate the transcriptional differences between difference CNV clones. WMA smoothing appeared in Figures, which is not described either.

Response: Thank you for the detailed comments. In the original manuscript, we structured the Methods sections for their functional purpose, so you can find the descriptions of WMA and KNN smoothing in section 4.4 for visualization, and KNN and HMM smoothing in section 4.7 for CNV states determination (Yes, KNN has been used in both parts). Nevertheless, these descriptions were kept in a concise format; in this revision, we have extended the description of them in the

above sections and added them explicitly into the updated Figure 1 and updated Figure S1g for illustrating HMM.

Besides the detailed methods section, we also briefly summarize these smoothing methods here. First, HMM is only used for CNV state determination. Essentially, once the CNV states for each gene (or gene bin) in each cell is calculated by the probabilistic model in Equations 2 & 4 or Equations 3 & 5 (as emission probability), we can smooth them by considering their neighboring genes on chromosome coordinate (as shown in the updated Figure S1g. and extended section 4.7, and Fig. R1 below). Second, KNN has been used both in visualization (section 4.4) and the CNV states determination (section 4.7). In both scenarios, the KNN provides a cell-cell similarity weight matrix M (see Eq. 1), which can be *left* multiplied to any cell-based features for a weighted-average smoothing for each cell feature independently (e.g., the BAF & RDR signal for visualization and CNV states probability). Third, WMA is applied in a similar manner but across genes, hence via a right matrix multiplication (Eq. 1). Now, we have further expanded the descriptions with more details about these three smoothing methods.

Regarding the potential circular logic in using KNN for smoothing, it indeed can be a concern, like detecting expression difference after grouping cells into subtypes. From this aspect, the KNN smoothing does assume that cells with similar transcriptome to have similar CNV profile. On the other hand, we carefully used such KNN smoothing, by 1) restricting it to a small neighborhood ($k=10$) and 2) emphasizing itself with a larger weight compared to its neighbors. Therefore, this smoothing can balance the denoising and retaining the cell diversity. For a similar reason, this strategy is also widely used in transcriptome smoothing, e.g., for unspliced RNAs in scVelo (Bergen et al 2020).

Figure R1. The transition of CNV states via HMM modelling within the RDR and BAF modules, respectively.

5. How CNV states are determined in XClone, as well as in other tools that are benchmarked?

The strategies need to be described in detail in addition of Fig. S2. The authors should not expect reader to understand the algorithm by throwing out a Figure panel. What are the definitions of copy loss A and copy loss B. It's confusing to use red to illustrate copy gain, and pink to show copy loss.

Response: Thank you for your valuable feedback. For the CNV determination, they have been explicitly described in Equations (Equations 2-3 for RDR, and Equations 4-5 for BAF). For other methods, we have now added more details on the methods (section 4.9; p. 17). With this comprehensive revision, we have also added more details and clarification on the CNV states detection in our method (see sections 4.4 to 4.7 and Fig. R2 for the graphical model presentation), especially for readers who are less familiar with statistical modelling.

For copy loss A and B, it refers to the loss of a certain genome region but from two different chromosome copies (paternal or maternal). In terms of color code, I hope the reviewer can understand that we are resolving a complex situation. In most cases, alleles A and B do not lose in the same tissue, where the most contrastive colors (dark blue and dark red) are used respectively for copy loss and gain. However, when both A and B allele losses happen (even though less common), we reserved high contrast colors, i.e., light blue and pink, to code these two alleles to attract more attention (you may refer to Fig. R3, our updated Supp. Fig. S2 on how the CNV states between RDR and BAF are combined with detailed color-coding selection).

Negative binomial modelling in RDR

Beta-binomial modelling in BAF

Figure R2. Probabilistic models within the RDR and BAF modules, respectively.

Figure R3. Updated Fig. S2 for the combine strategy illustration.

6. Is Fig.1f and Fig.S4a the same? Both are Numbat results of BCH869 data according to the legends.

Response: Yes, they are the same result of BCH869 dataset from Numbat. We highlight it in Main Figure 1f (now Fig. 2e) and display it again in Supplementary Figure S4a for easier comparison across different methods and convenient reading. We have clarified it in the legend of Fig. S4a. Thank you for bringing this to our attention.

7. Figure panels and labels need to be improved. Many fonts are too small to read.

Response: We have updated the figures accordingly.

8. How ROC values are calculated to compare between different tools? What are the ground truths?

Response: calculation of ROC values (of the BCH869 dataset) has been elaborated in section 4.9 and supplementary methods (Benchmarking by using ROC curve). The ground truth of the BCH869 dataset was well characterized by WGS and scRNA-seq from the original paper (Filbin et al, Science, 2018), which is shown in Fig. 1b (now Fig. 2a), and the details are listed in the Supp. Table S2. Also, we strive to ensure that our benchmarking is both fair and comprehensive.

By 'fair', we mean that we compare different tools or methods under the same conditions, using the same datasets (same genes and same cells) and evaluation metrics. We also ensure that each tool or method is used correctly, according to its documentation, and that any parameters are chosen in a manner that is appropriate and unbiased.

By 'comprehensive', we mean that our benchmarking includes a wide range of scenarios and data types. We aim to cover different types of CNVs, different levels of noise and complexity, and different sizes and types of datasets. This helps to ensure that our benchmarking results are representative and that they provide useful information for a wide range of potential users.

We believe that this approach to benchmarking helps to provide accurate, reliable, and useful

information for researchers and other users.

9. Fig.1f, what's the evidence of stating Numbat had a flow in missing copy loss on Chr14 (Line 89), instead of false discovery of Xclone in Fig.1e? Again, how are copy loss A and B decided? Numbat indeed detected the deletion.

Response: As mentioned above in point 8, the ground truth of BCH869 was listed in Fig. 1b (now Fig. 2a) and Supp Table S2, adapted from Fig. 3I-L and also Table S7 of the original paper (Filbin et al, Science, 2018). In the original paper, expression level analysis showed that the copy loss in chr14 in tumor cells, and analysis of haplotype frequencies revealed two subclones with different haplotypes, each of which has lost one distinct allele of chromosome 14 (J). This annotation of clone specific allelic loss was also well supported by the whole genome sequencing (WGS) on this sample (see Figures R4-5 below, kindly shared by the author).

The result from XClone is matched with the original papers', while Numbat only detected the deletion in one subclone and the other subclone with copy neutral in chr14 (original Fig. 1f & now Fig. 2e). Clearly, Numbat's result is discordant with the high purity copy loss reported from the WGS (Fig. R5), and also other methods, including InferCNV (Fig S4.b) for the deletion of chr14 in the whole cell population. All evidence suggests an erroneous result from Numbat, which possibly be resulted from the strategy of pseudo bulk clonal analysis in Numbat.

Figure R4. Clone specific allele loss supported by haplotype allele frequencies. Figure can be found in the original Fig. 3I-L of Filbin et al, Science, 2018.

Figure R5. Copy number analysis from the bulk whole genome sequencing in BCH869 cancer sample, with the total copy number shown in the top panel and the allelic fraction in the bottom panel. Focusing on **chr14**, there is a clear loss of absolute number with high purity, while the allelic fraction is only moderately deviated from 0.5, implying that the two alleles are present in the cell population, instead of only one allele. In other words, there are cells with loss of different alleles, well matching the two clones with respectively allele A loss and allele B loss, as reported

by the original study with WGS and uniquely detected by XClone from scRNA-seq. Figure kindly shared by the original authors of this study (Filbin et al, Science, 2018).

10. This is confusing again. Is Fig. S5 same as Fig.1g, h, i? The benchmarking at chromosomal arm scales seems unfair to other tools that may have higher resolution. Is same genomic reference version used to run all tools?

Response: No, they are different. Fig. 1 g,h,i is the benchmarking performance for each method at the gene scale, while Fig. S5 is at the chromosomal arm scale. As clarified above (point 2), XClone achieved a CNV resolution at the gene scale, while for fair comparison, we performed the benchmarking (ROC calculation) for both scales (Fig.1g, h, i for gene scale and Fig. S5 for arm scale). We have further clarified in the figure caption.

Yes, all tools used the same genomic reference version, i.e., GRCh37/hg19 (Genome Reference Consortium human build 37) on the BCH869 dataset, and GRCh38/hg38 on ATC2 and TNBC1 datasets. We chose the genome version for each dataset according to the downloaded bam files.

11. In ATC2 data, the major subclonal CNVs in Chr5 are detected in other tools, which is missed by XClone. As such, more datasets are suggested to test Xclone performance. The authors showed 5 panels of XClone results (Fig 2b-e and others). Which is the final result of XClone?

Response: Thanks for spotting the chr5 copy gain in clone 2 in ATC2. We agree that the CNV on chr5 is indeed clone 2 specific, particularly in terms of the absolute copy number as also indicated in the smoothed RDR signal in XClone (original Fig. 2c; new Fig. 3c). For such reason, we keep the two smoothed signals (RDR in panels c and BAF in panel e) and the detected CNV states for RDR (b), BAF (d) and combined (f). In general, the combined CNV state will be taken as the final result. On the other hand, we provide smoothed signals for both RDR and BAF as a double confirmation to ensure the robustness of our method and to avoid over interpretation of the CNV profiles.

We agree assessing more datasets will further strengthen the reliability of our method. Therefore, we have included a new dataset on astrocytoma from the scOne-seq paper (Yu et al, Sci Adv 2023), as described in Results section 2.5 and Figure 5. Briefly, this dataset has reliable ground truth at individual cell level thanks to the paired RNA and DNA sequencing technology. Impressively, our method has confidently identified the full CNV profiles of the minor CNV clone with astrocytes-like phenotype from the 10x Genomic snRNA-seq data.

12. What's the difference between Fig. 3c and Fig. S9, the last panel? Fig 3f and Fig. S10a?

Response: Fig. 3c and Fig. S9, the last panel are the same, and Fig 3f and Fig. S10a are the same as well. We presented them again in the supp file for easier reading, similar to Fig. 1f as mentioned in point 6. We have clarified the legends of the Supp figures.

13. XClone detected CNVs in normal cells, are they real or false calling? I'd suggest to test Xclone on scRNAseq data of some diploid tumors and normal organs to test the false calling and sensitivity.

Response: Thanks for the constructive suggestions. It is indeed important to balance the signal retaining and denoising. To demonstrate that XClone has reasonable level of noise in diploid cells, we have calculated the proportion of non-normal state in each diploid cell for both ATC2 and TNBC1 samples. We found this proportion is quite small for both datasets (4.0% on average; Fig.

R6 below). Since Numbat did not provide results for normal cells, we have chosen inferCNV for comparison in this context.

As a comparison, we found InferCNV gives 26.9% and 38.0% respectively for these two datasets for the proportion of genes out of [0.95, 1.05] region of transformed expression. Notably, this region cutoff gives a clear separation from the empirical distribution. Nonetheless, if using a more stringent cutoff of [0.9, 1.1], InferCNV still returns 4.2% and 9.2% of genes out of this region. In summary, we think our method keeps the false positive signals at a minimal level and generally will not lead to false discovery.

Figure R6. Noise level in diploid cells, detected by XClone (top) and InferCNV (bottom 2) on ATC2 (left) and TNBC2 (right) samples. Shown is the proportion of genes with non-normal state (by max state probability in XClone or transformed expression region in InferCNV) for each cell and their distribution across all normal cells.

14. What's the main advantages of XClone comparing to Numbat, in terms of CNV clone discovery? It seems all tools except for CasPER are able to define clones.

Response: Thank you for your question. XClone and Numbat are both effective tools for discovering clones defined by Copy Number Variations (CNVs), but there are several key advantages to using XClone:

1. Comprehensive output is flexible to do downstream analysis for users (we also get the same comment and feedback from XClone github issues). XClone outputs the comprehensive CNV probability matrix (Gene*Cell*CNVstate) for users to detect the clones, while Numbat only outputs the information of the regions identified as CNV. XClone can output independent BAF information for review but Numbat only outputs integrated information.
2. Robustness to complex clones' structure: XClone's algorithms are less sensitive to complex clones' structure compared to Numbat, thanks to XClone's high resolution at gene (bin) and individual cell level. Numbat mentioned that it exploits the evolutionary relationships between subclones to iteratively infer single cell copy number profiles and tumor clonal phylogeny. If the phylogeny is wrongly estimated, then the whole inferred CNV profile is affected (especially in BCH869 and ATC2). XClone focuses inferring the CNV from single cell, which may have background noise but can be denoised.
3. The whole preprocessing xcltk (upgraded from cellsnp-lite) is customized for XClone, and if there is matched genotyping from DNA, can be used in xcltk for more accurate allele-specific counting and be beneficial for clone detection. For example, the allele specific CN loss detection in BCH869 is helpful in subclone detection.

For these unique features, XClone allows for more accurate and robust clonal CNV detection (as shown in all BCH869, ATC2, TNBC1 and the new Gliomas sample), hence enabling clone discovery in a broader range of scenarios, including allele specific CN loss detection.

We appreciate your interest and hope this clarifies the advantages of XClone for CNV clone discovery.

15. Beyond the benchmarking results, what is the unique algorithmic strength/features of Xclone comparing to Numbat?

Response: In addition to the robustness and wide applicability described in point 14, XClone is also fine-tuned in its preprocessing steps, including (1) optimizing cellsnp-lite (implemented by our group and used by Numbat in its preprocessing) to avoid the double counting of UMIs in the upgraded xcltk (see the new Fig. 1b and also below Fig. R7), and (2) utilizing an optimized three-step BAF phasing (inspired from CHISEL) to account for the high noise in the raw BAF signals (Numbat only use the population phasing), which is described in detail in section 4.3.

Figure R7. The xcltk toolkit overview. It is also included in main Figure 1b.

16. How about the technical performance of XClone?

Response: We have demonstrated the superior technical performance of XClone, in terms of ROC values (in **Figure 1(g-i)** and **Supplementary Figure S5, 6**), and computational efficiency (computational complexity, memory requirements in **Supplementary Table S4**). To assist readers in understanding the strengths and weaknesses of the benchmarked methods, we have included **Supplementary Table S5** that summarizes the main features of each method. The table provides details on aspects such as the method's functions/modules, flexibility in use, efficiency, accuracy and other significant characteristics. The table is designed to be a quick reference guide for users to compare the different methods and choose the one that best suits their specific needs. We believe that this table, together with the detailed analysis presented in the manuscript, provides a comprehensive overview of the benchmarked methods.

Reviewer #2 (Remarks to the Author):

XClone adds a valuable resource for allele-specific CNV analysis, consistently outperforming its peers in rigorous comparative assessments. My concern revolves around its three operational modes (RAF, RDR, and combined): which one would be the optimal choice for different scenarios, and whether the tool can automatically pick the right one based on users' needs? The manuscript references "Supp. Table S5," which should provide a comprehensive analysis; however, upon review of the Supplementary Materials, this table appears to be missing.

Response: Thank you for the appreciation of the superior performance of our work and your insightful question. It has prompted us to think more about improving our tool tutorials and description of different modes for users. In short, for the result of XClone, the combined CNV states results should be used, while the BAF/RDR smoothing visualizations could be a good reference for comprehensive review. Following this, we elaborated on why the three modules of XClone can be used independently and provide detailed scenarios for the use of different modules.

XClone's three operational modes (BAF, RDR, and combined) are designed to provide flexibility depending on the data available and the specific goals of users' analysis.

1. RDR mode: This mode uses read depth ratio (RDR) information alone to detect CNVs. This might be the optimal choice when users only have expression count matrix available. Normal cells used as a reference is required.

2. BAF mode: This mode uses B-allele frequency (BAF) information alone to detect allele bias information in each cell for each gene. It might be the optimal choice when the raw sequencing data is available, and users are specifically interested in allelic imbalances or when users want to double-confirm the CNV states detected from expression information alone in many methods (inferCNV, CopyKAT). The RDR analysis relies on the normal cells used as a reference, thus if the genotype of these reference cells is not an ideal diploid state, then the analysis results using RDR may be affected. On the other hand, the BAF analysis is self-normalized and has a theoretical neutral value (0.5) for comparison when reference cells are limited, which makes BAF more stable to a certain extent and better reflects the actual changes in genotype. It's worth noting that phased BAF data often provide more accurate information, as 3-step phasing in XClone can distinguish different alleles at the same genetic locus, which helps accurately determine the changes of haplotype-aware allele proportions (for example, Fig. R8 shows BAF visualization before and after phasing and smoothing generated in BCH869, which is obvious to show the function of 3-step phasing).

Figure R8. The 3-step phasing in XClone in BCH869 dataset.

3. Combined mode: This mode uses both BAF and RDR information to detect CNVs. This would generally be the optimal choice when you have sequencing raw data (fastq/bam files) available. Combining the outputs of RDR and BAF module with appropriate combination strategy (Figure S2) can provide a more comprehensive view of the genomic state of each cell and can potentially lead to more accurate CNV and clone detection.

RDR/BAF mode operates independently, allowing the user to choose the most appropriate analysis strategy based on their specific needs and available data.

In general, whether using RDR or BAF, users need to handle the data carefully and understand the advantages and limitations of each method. Where possible, using combined mode may yield more comprehensive and accurate results (If users have raw sequencing data and follow the xcltk preprocessing and XClone pipeline, they will finally get all outputs and the combined results indicate the final CNV results).

The “Supp. Table S5” is available in the Excel file “442963_0_supp_7864780_ry4j44.xlsx”, but somehow manuscript submission system cannot convert it to the PDF file properly, so you may not find it in the PDF file.

We appreciate your thoughtful review and the time you've taken to help make our research better.

Reviewer #3 (Remarks to the Author):

This manuscript describes XClone, a novel method designed to enhance the identification of allele-specific Copy Number Alterations (CNAs) from single-cell RNA sequencing (scRNA-seq) data when compared to other recent methods. Given the abundance of scRNA-seq datasets in cancer research and the complexity of the problem, any advancements in CNA detection are relevant for cancer sequencing studies. However, this study fails to convincingly demonstrate that XClone adds a substantially novel methodological contribution surpassing previous methods or translates these improvements into the discovery of significantly novel biological insights.

Response: Thank you for the recognition of the urgent need for advancement in methodology for analyzing the CNVs in the abundant scRNA-seq across multiple cancer types. On the other hand, we would like to clarify that our work has made substantial contributions in both methodology improvement and biological insights, particularly in avoiding the catastrophic failure as Numbat did in two datasets (including one demo dataset in the Numbat, the ATC2 data).

Briefly, in this revision, we further clarified that the three key methodology contributions: 1) a tailored allelic read counting software xcltk to avoid UMI double counting in cellSNP-lite that is used in Numbat. As the team who developed cellSNP-lite, we highlighted the importance to avoid such reads double counting. 2) Although the BAF phasing part is indeed inspired by CHISEL, we extended it by warm initialization to avoid local optima which is crucial in our scenario as the expression between genes can be largely different. 3) We introduced a two-view graph-based smoothing (kNN graph across cells and neighbor gene graph), which is critically important as we aim to preserve the resolution at both gene (or 100-gene bin in BAF only) and cell level. A similar philosophy of preserving high resolution is also used in CHISEL and InferCNV, which are important to achieve high robustness.

For the biological insights, we have now added more analyses, especially on the clonal allelic loss on BCH869 sample where the allelic genotype affecting the expression of multiple cancer critical genes on chr14 (new Figure 2). Similarly, we found the widespread LOHs in TNBC1 sample also affect cancer-related genes probably via a regulatory role (new Figure 4). Lastly, we further added one more dataset on astrocytoma where our method confidently identifies a small CNV clone with normal astrocyte-like phenotype and indicated the whole genome duplication which is not supported by any other methods.

Major Concerns:

1. Lack of Novelty: While XClone does seem to offer some technical improvements over existing methods for identifying allele-specific CNAs from scRNA-seq data, it's important to acknowledge that it is not the pioneering method in this domain. Previous methods, such as Numbat, have pursued similar goals. It is somewhat unclear what novel concepts or features underpin XClone and how these innovations are expected in principle to translate into superior performance compared to Numbat or other similar previous methods. For instance, the principal novel feature of XClone, involving the computation of the BAF signal, appears to be an adaptation of the previous CHISEL method, originally developed for addressing a similar issue in scDNA-seq data. Consequently, XClone seems more like an incremental enhancement of previous methods rather than a substantial novel approach. Consideration of a more technical or algorithmic audience may

be apt for this contribution.

Response: Thank you for your insightful question. You are right that XClone and Numbat indeed have similar goals and share some features in their framework, e.g., both methods integrate BAF and RDR signals for CNV analysis, but we disagree that it is only an incremental enhancement. In this revision, we further clarified several mythology contributions of XClone: (1) detailed optimization to make the model robust, including optimizing cellsnp-lite to address the issue of double-counting UMIs and adapting three-step BAF phasing from CHISEL in pre-processing steps. (2) aim for single-cell resolution, instead of clone level in Numbat, avoiding making over-confident discovery. (3) XClone uses soft sharing / smoothing, while Numbat uses hard sharing between cells. (4) The separation and combination of RDR and BAF allows for indicating whole genome duplication as shown in the newly added astrocytoma data. Of note, our method has indeed been inspired from multiple aspects by CHISEL, an excellent tool for CNV analysis from scDNA-seq, while we have introduced various optimizations (as mentioned above) to ensure our method is tailored for scRNA-seq analysis to achieve high robustness and accuracy. We have also added these details into the expanded manuscript, especially the newly added Results sections 2.1 “High-level description of XClone model and workflow” and 2.5 about the new astrocytoma sample.

2. Insufficient Benchmarking: The absence of a formal benchmark employing a well-established ground truth prevents the demonstration of XClone's improved performance. The provided benchmark relies on a dataset featuring both scRNA-seq and whole-genome sequencing (WGS) data, but it does not establish an authentic ground truth; rather, it relies on inferred results from other methods. Additionally, WGS is not a suitable benchmark for single-cell results, as it is a bulk approach. To validate the method, a robust benchmark with a genuine ground truth is essential. I would recommend the authors to consider using either: (a) existing simulations, as done in [<https://doi.org/10.1101/2023.04.01.535197>] which used scRNA-seq simulator SPARSim [<https://doi.org/10.1093/bioinformatics/btz752>] and added simulated CNAs; or (b) using existing datasets having both scDNA-seq and scRNA-seq data (for example [<https://doi.org/10.1093/nargab/lqaa016>]).

Response:

We agree that the ground truth is essential for benchmarking methods. However, we would like to clarify there are multiple levels of ground truth.

First, bulk WGS gives the ground truth of CNV profiles across CNV clones but not in single cell, as the reviewer pointed out. In other words, from the genomic view, WGS-based ground truth is highly resolved and reliable, while from the cell view, it has extremely low resolution only at clone level. Nevertheless, if a method cannot even get the clone-level CNVs, its cell-level CNVs are meaningless. For example, in the BCH869 dataset, Numbat missed the copy loss on chr14 on a different allele in clone 2; as this is clone-level CNV, the bulk WGS ground truth is highly reliable (see Fig. R4-5 for reviewer 1).

Second, scRNA-seq itself contains “silver-standard” ground truth of a small set of CNV states thanks to its self-normalized BAF. For example, in the ATC2 dataset, from the smoothed BAF, we can rule out the ubiquitous copy loss as predicted by Numbat (Figure 2i), because if these were genuine copy loss, we would see strong BAF deviation (close to 0 or 1). Although such self-support ground truth is only applicable to distinguish very limited CNV states, they should be taken seriously as having reasonable reliability.

Third, scDNA-seq from unpaired cells gives ground truth with both reliable CNVs at genomic view and high resolution at cell level. However, the mapping between cells from scDNA-seq and scRNA-seq is non-trivial, making it only moderately better than using bulk WGS. Therefore, we did not try the dataset that the author suggested in (b), instead we used another dataset (see next).

Fourth, scDNA-seq from paired cells of scRNA-seq can give an ideal ground truth both reliable at the genomic view and highly resolved at the cell level. Therefore, we requested a data set from the recently published scOne-seq protocol, where scDNA and scRNA are from the same cell. Since the scRNA in scOne-seq is well-based and highly covered, we chose to use the 10x Genomics scRNA-seq from the same sample, which is more challenging but widely used. Also, mapping the 10x Genomics scRNA-seq to the scDNA-seq is more straightforward giving the anchoring scRNA-seq from scOne-seq (as shown by the data integration method in the scOne-seq paper, and in our new Fig. 5a). Therefore, we think this would be the most reliable ground truth one may be able to get so far. In our new Figure 5 on an astrocytoma sample, we demonstrated that our method is more powerful in detecting the minor CNV clone with normal astrocyte phenotype. Notably, our method uniquely indicates the whole genome duplication from this droplet-based scRNA-seq.

Fifth, simulated data is indeed often the first choice of ground truth as a necessary but not sufficient condition. However, simulation for clonal BAF is extremely challenging and the suggested SPARSim in (a) does not support this option, unsurprisingly.

Overall, we hope the reviewer agreed that we have utilized the most comprehensive ground truth, and they are sufficient to support the remarkable improvement in the robustness and accuracy in XClone compared to Numbat.

3. Absence of Novel Biological Insights: When XClone is applied to existing datasets, it appears to produce results largely akin to those generated by at least one of the other existing methods. It does not seem to yield novel biological insights, and the different results obtained between different methods largely appear relatively minor. This also raises concerns about which inference might align more closely with the truth, especially in the absence of orthogonal analyses.

Response: Thanks for the comments. We agreed that adding more results on biological insights will further enhance the demonstration of our method's usefulness. From our clarification in point 2, we hope that the reviewer agreed that the results we presented in the manuscript reflect the highly reliable ground truth (the points we emphasized). In other words, our method can uniquely detect clonal CNV, for example the allelic loss on the BCH869 sample and widespread LOH in TNBC1 sample.

From such unique detection, we aimed to identify its potential impact on the transcription changes. Therefore, we performed differential expression (DE) analysis between the two allelic clones in BCH869 with a focus on the genes on chr14. Interestingly we found 9 significant DE genes between the two alleles where 4 are cancer related genes, including RPS6KA5, PYGL, FUT8, and INF2. Similarly, we performed the DE analysis on the TNBC1 sample by comparing each tumor clone to the diploid clone and found a substantially increased proportion of the DE genes in the LOH regions compared to other regions. Also, these clone 1 ungraduated DE genes show

a strong enrichment in the Wnt signaling pathways, well explaining the uncontrolled cell proliferation.

Furthermore, as mentioned in the point 2 above, we have added new dataset on an astrocytoma sample, where our method not only accurately detect the CNVs with the newly emerged clone in the second recurrence, but uniquely indicates the whole genome duplication. These discoveries can potentially link to various biological insights, both for understanding somatic evolution and also therapeutic targeting clones.

4. Clarity of Text: The manuscript offers limited explanations and is heavily laden with technical terminology, making it challenging to follow, particularly for a non-expert audience. A dedicated Results section that elucidates the novel approach and its unique attributes is also missing, hindering a clear understanding of the anticipated advantages over previous methods and which elements are genuinely innovative versus adaptations from prior approaches. Lastly, the manuscript's brevity leaves room for additional details and explanations. Furthermore, certain methodological sections lack proper mathematical descriptions, further complicating comprehension. For example, one of the main steps of XClone in Section 4.3 lacks any equation or formal mathematical description. Also what are the details of the defined HMM (covariates, model, etc.)? And several other important details are missing.

Response: Thanks for these constructive comments. Since this work contains rich aspects (two modalities, each with pre-processing, visualization and CNV states estimation), we aimed to keep it concise to avoid unnecessarily lengthy otherwise. Nevertheless, we do agree with the reviewer that more details can further enhance the clarity of our work.

Therefore, in this revision, we have substantially expanded the manuscript in both results and methods. Major expansions cover 1) adding a results section for a high-level description of XClone, including unique attributes compared to other methods and details of the methodology contributions; 2) more details to the Methods section, including elaborating the BAF preprocessing in section 4.3 and expanding the details in HMM. Since the HMM model is a generic framework for smoothing the clustering in a 1D structure, it is shared by both BAF and RDR modules; therefore, we prefer to keep it in a separate subsection and hope the reviewer will appreciate this brevity.

Minor Terminology Comment:

The manuscript utilizes the term "Copy Number Variants (CNVs)" incorrectly in this context. CNVs are typically associated with germline variants. For somatic alterations in cancer cells, "Copy Number Aberrations/Alterations (CNAs)" is the standard and appropriate terminology.

Response: Thanks. As the community broadly uses both CNV and CNA for somatic alternations (CNV: inferCNV, CaSpER, CNV; CNA: CopyKat, CHISEL), we would like to simply keep using CNV. Of note, we term "copy number variations" for CNV (not "variants"); when we use the term "variants", we mean germline SNP. This terminology is used throughout the manuscript.

References

- [1] Filbin, M.G., Tirosh, I., Hovestadt, V., Shaw, M.L., Escalante, L.E., Mathewson, N.D., Neftel, C., Frank, N., Pelton, K., Hebert, C.M., et al. Developmental and oncogenic programs in h3k27m gliomas dissected by single-cell rna-seq. *Science* **360**(6386), 331–335 (2018).
- [2] Yu L, Wang X, Mu Q, Tam SS, Loi DS, Chan AK, Poon WS, Ng HK, Chan DT, Wang J, Wu AR. scONE-seq: A single-cell multi-omics method enables simultaneous dissection of phenotype and genotype heterogeneity from frozen tumors. *Science Advances*. 2023 Jan 4;9(1): eabp8901.

Reviewers' comments:

Reviewer #1 (Remarks to the Author):

Reviewer #2 (Remarks to the Author):

The authors have adequately addressed my comments, which I appreciate. I have no additional concerns.

Reviewer #2 (Remarks on code availability):

The code is well-documented via GitHub. I didn't run their code and therefore couldn't comment on the reproducibility.

Reviewer #3 (Remarks to the Author):

The authors deserve commendation for the substantial additional work, which has significantly enhanced the clarity of the study and provided stronger support for the reported results. However, a major concern persists, demanding further attention: the necessity for a comprehensive benchmark with a robust ground truth. This is particularly crucial given the plethora of alternative methods available to address related tasks. The scientific community requires guidance on discerning the contexts and settings in which XClone should be preferred over previous tools. Moreover, it is important to explore the limitations of XClone and its advancements compared to previous tools when considering variations in different parameters such as the number of cells, levels of aneuploidy, sequencing coverage, etc.

While the introduction of the new scOne-seq dataset is sufficient for showcasing the novel impact of XClone, a single dataset is not sufficient for investigating diverse settings. To address this, I still recommend considering the use of simulations. While developing new sophisticated simulations for allelic copy numbers in scRNA is non-trivial, recent studies have successfully proposed and applied simplistic alternatives that could be viable in this case. For instance, the read counts generated by existing simulators like SPARSim can be scaled by simulated total copy numbers (as done in recent scRNA copy number studies like <https://doi.org/10.1371/journal.pcbi.1011557>) and BAFs can be simulated using binomial models based on the total number of simulated counts and the frequency of allele copy numbers.

Lastly, I would still invite the authors to reconsider the choice of terminology CNVs vs CNAs, as the current usage is incorrect in biology. CNVs is a well-established terminology indicating germline events that are inherited and vary across individuals of a population. On the other hand, CNAs is used to differentiate somatic events that occur in somatic cells within the same individual (see an example of the distinction in <https://doi.org/10.1038/nature10983>).

Reviewer #3 (Remarks on code availability):

At this stage my review of the code was only brief, but the available repository seems to be quite appropriate and it contains quite detailed descriptions of the code, instructions of use, demos, etc.

Point-by-point response

Reviewer 1: page 2.

Reviewer 2: page 2.

Reviewer 3: pages 2 to 7.

The responses are highlighted in blue.

Reviewer #1 (Remarks to the Author):

Response:

Thank you for your time and effort in reviewing our manuscript. In this revision, we have further implemented a unique simulator (scCNASimulator) to support allelic reads simulation. Therefore, we generated simulated data with solid ground truth and performed comprehensive simulated analyses. These new simulated results consistently support our claim that in complex scenarios (e.g., clone specific allelic loss) our method is highly robust and achieves substantial improvement compared to existing methods. On the other hand, in relatively easy (or clean) settings, both existing methods like Numbat and our XClone are also good choices.

Hope these revisions have addressed your concerns.

Reviewer #2 (Remarks to the Author):

The authors have adequately addressed my comments, which I appreciate. I have no additional concerns.

Reviewer #2 (Remarks on code availability):

The code is well-documented via GitHub. I didn't run their code and therefore couldn't comment on the reproducibility.

Response:

We are grateful for your positive feedback and are pleased to hear that your concerns have been fully addressed. Thank you once again for your time and valuable input.

Reviewer #3 (Remarks to the Author):

The authors deserve commendation for the substantial additional work, which has significantly enhanced the clarity of the study and provided stronger support for the reported results. However, a major concern persists, demanding further attention: the necessity for a comprehensive benchmark with a robust ground truth. This is particularly crucial given the plethora of alternative methods available to address related tasks. The scientific community requires guidance on discerning the contexts and settings in which XClone should be preferred over previous tools. Moreover, it is important to explore the limitations of XClone and its advancements compared to previous tools when considering variations in different parameters such as the number of cells, levels of aneuploidy, sequencing coverage, etc.

While the introduction of the new scOne-seq dataset is sufficient for showcasing the novel impact of XClone, a single dataset is not sufficient for investigating diverse settings. To address this, I still recommend considering the use simulations. While developing new sophisticated simulations for allelic copy numbers in scRNA is non-trivial, recent studies have successfully proposed and applied simplistic alternatives that could be viable in this case. For instance, the read counts

generated by existing simulators like SPARSim can be scaled by simulated total copy numbers (as done in recent scRNA copy number studies like <https://doi.org/10.1371/journal.pcbi.1011557>) and BAFs can be simulated using binomial models based on the total number of simulated counts and the frequency of allele copy numbers.

Lastly, I would still invite the authors to reconsider the choice of terminology CNVs vs CNAs, as the current usage is incorrect in biology. CNVs is a well-established terminology indicating germline events that are inherited and vary across individuals of a population. On the other hand, CNAs is used to differentiate somatic events that occur in somatic cells within the same individual (see an example of the distinction in <https://doi.org/10.1038/nature10983>).

Reviewer #3 (Remarks on code availability):

At this stage my review of the code was only brief, but the available repository seems to be quite appropriate and it contains quite detailed descriptions of the code, instructions of use, demos, etc.

Response:

Thanks for the further clarification on the terminology, and we agreed it is good to keep somatic and germline CNVs in different terms and we have changed the CNV to CNA throughout.

Also thank you for your thorough evaluation and constructive feedback. We acknowledge the importance of a benchmark with robust ground truth to validate the performance of XClone comprehensively, therefore developed a simulator for this purpose under various conditions. Based on your recommendations, we have made major changes as follows:

1. We designed and implemented a simulator, scCNASimulator to generate allelic reads.

As replied in our previous revision, the existing simulators, including the suggested methods SPARSim and beta-binomial, can only generate count matrices but not sequencing reads. Not to mention that there is no tool supporting phased alleles (haplotype) covering multiple SNPs. However, phased alleles are major information for XClone (and Numbat) and we do need reads/bam file to run Numbat for comparison. For these reasons, we had to give up on performing simulations in the previous revision. On the other hand, we did seriously consider the reviewer's suggestions and have made tremendous efforts (in the past half year) to develop a new simulator from scratch to support haplotype aware reads data.

The details of the simulator can be found in the new **Supplementary Technical Notes** (Supplementary file). Briefly, this tool is a Python package designed for CNA simulation in droplet-based scRNA-seq data. It mainly takes an indexed BAM file and a clone-specific CNA profile as input and outputs a new indexed BAM file containing re-sampled reads with new tags (in UMIs) for the desired CNA alignments. Critically, when performing allele drop, it will use the genotyping phasing information to drop a specific allele. We have added the cartoon below (Figure R1; same as main Figure 6a).

Figure R1. A tailored simulator was developed to create scRNA-seq reads that include allele-specific copy number loss within selected genomic regions. Take allele-specific copy loss on chromosome 3 in 2 subclones as an illustrative scenario: the left panel displays the original, unprocessed data from various cells spanning genomic regions. The right panel shows that tumor cell 1 is assigned to subclone 1, which exhibits a loss of the REF allele in certain regions of chromosome 3, while tumor cell 2 is part of subclone 2, characterized by a loss of the ALT allele. Cell 3 retains the initial, unaltered state.

2. We performed a series of simulations to mimic common challenging scenarios of CNA detection.

To simulate high-fidelity reads and CNAs, our scCNASimulator generally requires a reliable seed dataset (in BAM format). To achieve this, a fresh gastric cancer tissue was sequenced via 10x Genomics 5' scRNA-seq kit (GX109-T1c), and generously shared by our collaborators (the newly added co-authors). This sample is ideal as a seed dataset for the simulations: 1) the CNA clonal structure is simple with all microenvironment cells as CNA neutral and tumor cells as one CNA clone; 2) the CNA profile of the tumor cells is readily detectable and covers gain, loss and LOH, both of which are consistently supported by bulk whole exome-seq, scDNA-seq and scRNA-seq data on this tissue (Supp. Fig. S15-17). The CNA profile of the input seed dataset is also shown below (Figure R2a; same as Supp. Fig. S18a), including copy gain at chr8, chr20, copy loss at chr12p (chr12:7288360-16757762) and chr18, and CN-neutral LOH at chr1q, chr6p, chr10q and chr11q.

GX109-T1c Simulation: Ground truth

Figure R2. Illustration of the CNA ground truth for different simulation scenarios. (a) The CNA ground truth of the GX109-T1c scRNA-seq dataset. (b) The ground truth for the simulation scenario: sub-clonal division within the tumor populations via allele-specific copy loss. (c) The ground truth for simulation scenario: reference cells down sampling to minor quantities. (d) The CNA ground truth for simulation scenario: tumor cells down sampling to limited quantities.

To further evaluate the performance of CNA detection in different situations, a series of *in silico* experiments were performed within the GX109-T1c dataset. These experiments were designed to mimic more intricate clonal architectures and a pronounced disproportion in the count of normal versus cancerous cells to assess the capability of the benchmarked methods in detecting CNAs robustly. The simulations cover the following three common challenging scenarios in six settings (ground truth profiles accordingly shown above in Figure R2b-d, same as Supp. Fig. S18b-d):

- **Simulation scenario 1:** Sub-clonal division within the tumor populations via allele-specific copy loss in chromosome 3 (2 subclones lose different allele).
- **Simulation scenario 2:** The whole dataset reference cells down sampling to extremely sparse minor quantities (e.g., 10 cells, 5 cells).
- **Simulation scenario 3:** The whole dataset tumor cells down sampling to extremely sparse minor quantities (e.g., 22 cells, 10 cells, 5 cells).

We incorporated simulated datasets to benchmark XClone against Numbat (or inferCNV if Numbat fails) and summarized the performance in new Figure 6 in the manuscript. Extending our validation to include multiple simulated datasets that cover a spectrum of conditions, we can better understand the limitations and strengths of XClone and provide clearer guidance to the

scientific community. **Supp Fig. S19-21** show the results from different CNA detection methods applied on the simulated datasets.

GX109-T1c allele-specific copy loss on chromosome3 Simulation

a. XClone BAF

b. XClone Combine

c. Numbat

d. inferCNV

Figure R3. (a-b) XClone shows good performance on the simulated allele-specific copy loss in chromosome 3, with visualization of the smoothed BAF signal (a) and called CNA states (b). (c-d) Numbat and inferCNV cannot show the expected CNA profile well on the simulated allele-specific copy loss in chromosome 3. Numbat cannot call the copy loss in chromosome 3, for both absolute number and the specific alleles. InferCNV can call the copy loss in chromosome 3 but no allele specific information and no LOH events detected as expected.

3. We provided a detailed guide for users and demonstrated the strengths of our approach.

With the above simulations, we have compared XClone with Numbat (or InferCNV if Numbat failed to detect or output) and compiled the results for users as a detailed guideline. The full results and descriptions are provided in the new results section, together with new Figure 6, Supp. Fig. S19-21 and Supp. Table S5.

Here, we summarize as these three guidelines:

- 1) When clone-specific allele loss exists (results shown Figure R3 above; same as Supp. Fig.

S19), XClone is capable of detecting both, while Numbat does not support this complex scenario and InferCNV can only detect the absolute copy loss ignoring the allele information and potential subclones (this mimics the BCH869 dataset).

- 2) When the tumor cells are too few, XClone remains capable of working, while Numbat fails to output anything. This may reflect the erroneous results in ATC2 data where the tumor cells are the minority. Of note, inferCNV shows good robustness for low tumor prevalence, if one is mainly concerned about the absolute number.
- 3) When the reference cells are as low as 5, both XClone and Numbat generally work well.

Overall, we have utilized the simulations for the most comprehensive ground truth, which show consistent results to our observations in experimental datasets that our XClone achieves remarkable improvement in the robustness and accuracy of XClone compared to Numbat, especially in complex scenarios.

REVIEWERS' COMMENTS

Reviewer #3 (Remarks to the Author):

The authors have done a large amount of high quality additional work that now fully addresses my concerns.

I believe that the new simulator and the related benchmark are now a very key and important contribution of this work, that should be highlighted in Abstract and Introduction, and it will be widely useful for the community.

Reviewer #3 (Remarks on code availability):

The software repository and related documentation are of very high quality. The tutorials are particularly well designed.

I would recommend the authors to consider adding XClone to Bioconda, to even further maximise the usability of this method (<https://bioconda.github.io/>). This should be straightforward given the tool is already in pip.

Point-by-point response

The responses are highlighted in blue.

Reviewer #3 (Remarks to the Author):

The authors have done a large amount of high quality additional work that now fully addresses my concerns.

I believe that the new simulator and the related benchmark are now a very key and important contribution of this work, that should be highlighted in Abstract and Introduction, and it will be widely useful for the community.

Reviewer #3 (Remarks on code availability):

The software repository and related documentation are of very high quality. The tutorials are particularly well designed.

I would recommend the authors to consider adding XClone to Bioconda, to even further maximise the usability of this method (<https://bioconda.github.io/>). This should be straightforward given the tool is already in pip.

Response:

We are grateful for your positive feedback and are pleased to hear that your concerns have been fully addressed. Thank you once again for your time and valuable input.

We have updated the Abstract and Introduction to mention the related benchmark and simulator. We will consider adding XClone to Bioconda later. Thank you for your suggestion to further maximise the usability of this method!